# Towards a Theoretical Understanding of In-context Learning: Stability and Non-I.I.D Generalisation

**Yingjie Wang**[1]    **Yutian Zhou**[2]    **Shi Fu**[1]    **Yuzhu Chen**[3]
**Yongcheng Jing**[1]    **Leszek Rutkowski**[4]    **Dacheng Tao**[1][*]

[1]Generative AI Lab, College of Computing and Data Science, Nanyang Technological University
[2]Fudan University
[3]University of Science and Technology of China
[4]The AGH University of Krakow
`{yingjiewang1201,dacheng.tao}@gmail.com, apollo.yt.zhou@gmail.com`
`shi011@e.ntu.edu.sg, cyzkrau@mail.ustc.edu.cn`
`yongcheng.jing@ntu.edu.sg, rutkowski@agh.edu.pl`

## Abstract

In-context learning (ICL) has demonstrated significant performance improvements in transformer-based large models. This study identifies two key factors influencing ICL generalisation under complex non-i.i.d. scenario: algorithmic stability and distributional discrepancy. First, we establish a stability bound for transformer-based models trained with mini-batch gradient descent, revealing how specific optimization configurations interact with the smoothness of the loss landscape to ensure the stability of non-linear Transformers. Next, we introduce a distribution-level discrepancy measure that highlights the importance of aligning the ICL prompt distribution with the training data distribution to achieve effective generalisation. Building on these insights, we derive a generalisation error bound for ICL with asymptotic convergence guarantees, which further reveals that token-wise prediction errors accumulate over time and even lead to generalisation collapse if the prediction length is not properly constrained. Finally, empirical evaluations are provided to validate our theoretical findings.

## 1 Introduction

In recent years, the AI community has witnessed the emergence of influential Large Models (LMs) such as Generative Pretrained Transformers (GPTs) (Brown et al., 2020; Achiam et al., 2023; Radford et al., 2018; 2019), LLaMa (Touvron et al., 2023), and Pathways Language Model (PaLM) (Chowdhery et al., 2023). A particularly attractive characteristic of LMs is their in-context learning (ICL) capability, which enables effective predictions on downstream tasks using only a short context, without requiring any parameter fine-tuning (Black et al., 2022).

Recently, the empirical success of ICL has attracted growing interest in theoretically analyzing its generalisation capability. Li et al. (2023) establish optimization-independent generalisation bounds for ICL under i.i.d. inputs or trajectories derived from dynamical systems. Other works incorporate training dynamics and prompt structure into the analysis, examining how architectures and optimization strategies influence ICL performance (Huang et al., 2024; Li et al., 2024a; Chen et al., 2024). Notably, Wu et al. (2024) establish a statistical task complexity bound for the attention model pretraining and indicates pretrained model closely matches the optimally tuned ridge regression by achieving nearly Bayes optimal risk on unseen tasks. However, these studies rely on simplifying data assumptions that limit their applicability to real-world settings, such as the pairwise orthogonal token pattern imposed by (Huang et al., 2024; Li et al., 2024a) and the independent token sampling assumption in (Chen et al., 2024; Wu et al., 2024).

---

[*]Corresponding author.

Table 1: Theoretical Contributions (✓-has the given information, ✗-hasn't the given information)

| | Multi-Head Multi-Layer | Generalisation Analysis | Optimization Dependent | Distribution Shift | No Special Input Structure | Orthogonality Free |
|---|---|---|---|---|---|---|
| Li et al. (2024b) | ✓ | ✗ | ✗ | ✓ | ✗ | ✓ |
| Feng et al. (2023) | ✓ | ✗ | ✗ | ✓ | ✓ | ✓ |
| Chen et al. (2024) | ✓ | ✓ | ✓ | ✗ | ✓ | ✗ |
| Bai et al. (2024) | ✓ | ✗ | ✗ | ✓ | ✗ | ✓ |
| Yang et al. (2024b) | ✓ | ✓ | ✓ | ✓ | ✗ | ✓ |
| Li et al. (2024a) | ✗ | ✓ | ✓ | ✓ | ✗ | ✗ |
| Ours | ✓ | ✓ | ✓ | ✓ | ✓ | ✓ |

**Special input structure** refers to prompts structured in a specific format to satisfy theoretical constraints.
**Orthogonality-free** refers to data that is not constrained by orthogonal patterns in its generation

This paper relaxes these ideal assumptions and theoretically analyzes the generalization of nonlinear Transformers for next-token prediction in ICL, leveraging algorithmic stability (Bousquet & Elisseeff, 2002; Charles & Papailiopoulos, 2017; Liu et al., 2017) and discrepancy measure (Kuznetsov & Mohri, 2015; 2020; Wang et al., 2022). Our main theoretical contributions are:

**Algorithmic Stability and Discrepancy Measure:** Algorithmic stability ensures that small changes in training data do not cause large inference variations. We theoretically identify conditions under which Transformers achieve stability under mini-batch gradient descent and quantify discrepancy across different scenarios. Theorem 1 reveals three key insights: 1) for a sufficiently smooth loss landscape, algorithmic stability is well-controlled, and allows iteration number to scale polynomially with the training sample size; 2) in non-smooth scenarios, stability deteriorates rapidly as iterations number increase, especially with a small learning rate, making it advisable to limit iterations number to a logarithmic scale relative to the sample size; 3) regardless of whether the landscape is sufficiently smooth, an appropriately chosen step size can ensure that the convergence rate of algorithmic stability achieves $O(N^{-1})$, where $N$ denotes the sample size. The discrepancy measure captures distribution shift between training and target data. To quantify this discrepancy, Theorems 2–3 establish a stability-dependent asymptotically vanishing bound for the i.i.d. case, and a bound based on sequential Rademacher complexity for the non-i.i.d. setting.

**Generalisation Bounds:** Theorem 4 establishes the generalisation error of Transformer-based models under ICL scheme by leveraging algorithmic stability and the discrepancy measure, revealing: 1) In the ideal i.i.d. data scenario, the ICL generalisation error achieves a convergence rate of $O(N^{-\frac{1}{2}})$ with appropriately chosen iteration number and batch size, regardless of the loss landscape's smoothness; 2) In the non-i.i.d. data scenario, effective generalisation requires properly weighting training samples and suitable ICL prompting, particularly when the loss landscape exhibits insufficient smoothness; 3) The generalization error accumulates across the intermediate tokens generated by the model. Theorem 5 suggests that, to ensure effective generalisation, the length of next-token predictions should be constrained to grow at most logarithmically with the sample size.

## 2 RELATED WORK

A major line of work investigates the approximation capabilities of ICL in solving diverse tasks, while another focuses on their generalization and dynamic training behavior, aiming to establish theoretical guarantees for adaptation to unseen tasks under i.i.d. and distribution shift settings. In the research line of approximation analysis, Akyürek et al. (2023); Bai et al. (2024) demonstrate that Transformers are expressive to conduct many machine learning algorithms in context, such as ridge regression and Lasso regression. Moreover, a series of studies prove the existence of Transformer architectures capable of implementing gradient-based methods and their variants when given appropriate prompts (Von Oswald et al., 2023; Ahn et al., 2023; Ding et al., 2024). Within a self-training framework, Fu et al. (2025) establish the generalization error of Transformers with ICL. A particularly influential subclass of ICL prompts, Chain-of-Thought (CoT), has been extensively studied as a structured form of in-context reasoning. Several works show that CoT-enhanced Transformers are strictly more expressive than their standard counterparts (Feng et al., 2023; Li et al., 2024c; Merrill & Sabharwal, 2023). Specifically, Malach (2024) prove that next-token predictors trained on CoT data can efficiently

simulate any Turing-computable function, while Li et al. (2024b) show that Transformers can even learn multi-layer perceptrons in context.

In another research line, Huang et al. (2024) explore the training dynamics and generalisation of ICL on single-attention Transformers. Huang et al. (2024) analyze the generalization properties of single-head attention Transformers, while Chen et al. (2024) study the gradient flow dynamics in multi-head architectures for multi-task linear regression. Further, Cui et al. (2024) and Yang et al. (2024a) provide theoretical evidence for the superiority of multi-head attention and standard Transformers over single-head and recurrent baselines in various reasoning settings. More recently, Gong et al. (2025) examine the emergence of ICL capabilities in autoregressive next-token prediction models through PAC-Bayes theory. In addition, Li et al. (2025) provide sample complexity and bounds for training Transformers to acquire CoT capabilities under a token orthogonality assumption. Recent theoretical studies have provided elegant geometric and optimization-based explanations of in-context learning through structured concept representations. These works substantially deepen the mechanistic understanding of how semantic geometry and task-vector behavior emerge in transformer models (Bu et al., 2024; 2025). Our work tackles a complementary question. Without assuming any latent concept geometry, we develop a distribution-shift-aware generalization framework via algorithmic stability and discrepancy.

To clearly highlight our contributions, Table 1 provides a comparative analysis of existing theoretical works, emphasizing key differences in assumptions and results. Unlike prior works that rely on restrictive assumptions, such as orthogonal token patterns (Li et al., 2024b; Feng et al., 2023) or i.i.d. sampling (Chen et al., 2024), our analysis does not require idealized input structures and explicitly handles non-i.i.d. settings with distribution shift. This makes our generalization bounds applicable to a wider range of realistic ICL scenarios, including those where training and inference environments differ significantly.

## 3 PROBLEM SETUP

Suppose we have a sample of size $N$, where the $i$-th sample variable is denoted as $(X^i, \mathbf{C}^i)$, with $X^i$ representing the query variable and $\mathbf{C}^i = (C_1^i, \ldots, C_{N_c}^i)$ representing the length-$N_c$ output sequence. Importantly, our theoretical results allow for these sample variables to follow distinct distributions.

A typical length-$N_p$ ICL prompt consists of an example set $D^i = \{(X^i, \mathbf{C}^i)\}_{i=1}^{N_e}$, which is contextually associated with the pair $(X^i, \mathbf{C}^i)$, followed by a query input $X^i$. We formally represent the prompt as $\mathbf{P}^i = [D^i, X^i]$, where $[D^i, X^i]$ denotes the concatenation of the example set $D^i$ and the query input $X^i$ into a single flattened input vector. In practice, we typically predict each intermediate token $C_j^i, j = 1, \cdots, N_c$, in an autoregressive manner, where the prompt for predicting the $j$-th token incorporates the token from the previous $j - 1$ steps. Accordingly, we denote the integrated prompt for $j$-th token prediction as $\mathbf{P}^{i,j} = [\mathbf{P}^i, C_1^i, \cdots, C_{j-1}^i]$.

In practice, instead of relying on the correct intermediate tokens, the estimated intermediate tokens are more commonly used to predict the next-token. Under this more general scenario, we define $\hat{\mathbf{P}}^{i,j} = (\hat{\mathbf{P}}^{i,j-1}, \mathcal{T}(\hat{\mathbf{P}}^{i,j-1}))$ with $\hat{\mathbf{P}}^{i,0} = \mathbf{P}^i$, where $\mathcal{T}$ is a Transformer-based model. However, this approach inevitably results in error accumulation. Appendix G establish the gap between the generalisation performance with $\hat{\mathbf{P}}^{i,j}$ and $\mathbf{P}^{i,j}$, highlighting the impact of these accumulated errors.

For convenient reference, Appendix A provides a summary of the notations used in this paper.

### 3.1 TRANSFORMERS ARCHITECTURE

This section introduces the widely adopted non-linear Transformer architecture, which comprises self-attention mechanisms and a multi-layer perceptron (MLP) module.

**Definition 1.** *(Multi-head Self-Attention Module) For any given length-$N_p$ prompt*

$$\mathbf{P} = \begin{bmatrix} - & z_1^T & - \\ - & z_2^T & - \\ \vdots & \vdots & \vdots \\ - & z_{N_p}^T & - \end{bmatrix} \in \mathbb{R}^{N_p \times D},$$

---

**Algorithm 1** Mini-batch Gradient Descent Optimizer for Transformer

---

**Input:** Observations $S = \{(\mathbf{p}^i, \mathbf{c}^i)\}_{i=1}^N$, Initialization $\theta^0$, Max-Iter $Q$, $q = 0$, Batch Size $|B|$.
**Output:** $\hat{\theta} = \theta^Q$.
**For:** $q \leq Q$;
   $q \leftarrow q + 1$;
   Stochastically Sampling $B \subset \{(\mathbf{p}^i, \mathbf{c}^i)\}_{i=1}^N$;
   $\theta^q = \theta^{q-1} - \frac{\eta_{q-1}}{|B|} \sum_{i \in B} \nabla_\theta \hat{\mathcal{L}}(\mathcal{T})$;

---

*suppose that there are $N_a$ attention module $\mathcal{A}(\cdot) : \mathbb{R}^{N_p \times D} \to \mathbb{R}^{N_p \times D}$, with parameters $O_m \in \mathbb{R}^{D \times D}$ and $\{(V_m, Q_m, K_m)\} \in \mathbb{R}^{D \times D}$ for each attention module $m = 1, \cdots, N_a$. The attention score associated with $i$-th token $(\mathcal{A}(\cdot))_{i,:} : \mathbb{R}^{N_p \times D} \to \mathbb{R}^{1 \times D}$ is given by*

$$\mathcal{A}(\mathbf{P})_{i,:} := \sum_{m=1}^{N_a} \left[ \sum_{j=1}^{N_p} \text{softmax}\left(z_i^T Q_m K_m z_j\right) z_j^T V_m \right] O_m,$$

where the softmax mapping is defined by

$$\text{softmax}(z_i^T Q_m K_m z_j) = \frac{e^{z_i^T Q_m K_m z_j}}{\sum_{j=1}^{N_p} e^{z_i^T Q_m K_m z_j}}.$$

The vector-based form can be derived easily:

$$\mathcal{A}(\mathbf{P}) := \begin{pmatrix} \mathcal{A}(\mathbf{P})_{1,:} \\ \vdots \\ \mathcal{A}(\mathbf{P})_{N_p,:} \end{pmatrix} \in \mathbb{R}^{N_p \times D} = \sum_{m=1}^{N_a} \text{softmax}\left(\mathbf{P} Q_m K_m \mathbf{P}^T\right) \mathbf{P} V_m O_m.$$

**Definition 2.** *(MLP Module) For any given matrix $\mathbf{Z} \in \mathbb{R}^{N_p \times D}$, a (token-wise) MLP layer with hidden dimension $D$ is denoted as $\mathcal{M}(\mathbf{Z}) = \text{ReLU}(\mathbf{Z}W_1)W_2 \in \mathbb{R}^{N_p \times D}$, where $W_1, W_2 \in \mathbb{R}^{D \times D}$ are parameters matrices.*

Given any prompt $\mathbf{P}$, we have the following inference process of $l$-layer Transformer

$$\mathbf{H}^l = T^l(\mathbf{H}^{l-1}) := \mathcal{M}^l\left(\mathcal{A}^l(\mathbf{H}^{l-1})\right), l = 1, ..., L,$$

where $\mathbf{H}^l$ is the output of $l$-layer block of Transformer and $\mathbf{H}^0 = \mathbf{P}$. Consequently, the Transformer architecture with $L$ layers can be expressed as $\mathcal{T}(\mathbf{P}) = T^L \circ T^{L-1} \circ \cdots \circ T^1(\mathbf{P})$. It is important to highlight that in typical usage, only the last token from the final layer, denoted as $\mathcal{T}(\mathbf{P})_{*,:}$, is utilized as the output corresponding to the queried response.

## 3.2 Training with Stochastic Gradient Descent

This paper considers a training process where each training example is aligned with the test setup. This learning scheme ensures that the model learns to mirror the inference process at test time. Furthermore, the empirical risk formulation employed in this work is also widely used in both theoretical analyses (Li et al., 2024a; Yang et al., 2024b), empirical studies from practical applications (Min et al., 2022), and dataset development Longpre et al. (2023).

Given $N$-size sample set $S = \{(\mathbf{p}^i, \mathbf{c}^i)\}_{i=1}^N$, the training objective is formulated as:

$$\hat{\mathcal{L}}(\mathcal{T}) = \sum_{i=1}^N \frac{q_i}{N_c} \sum_{j=1}^{N_c} \ell\left(\mathcal{T}(\mathbf{p}^{i,j-1})_{*,:}, c_j^i\right),$$

where $q_i, i = 1, ..., N$, represent the weights for the training data, reflecting its relative importance in the overall optimization process.

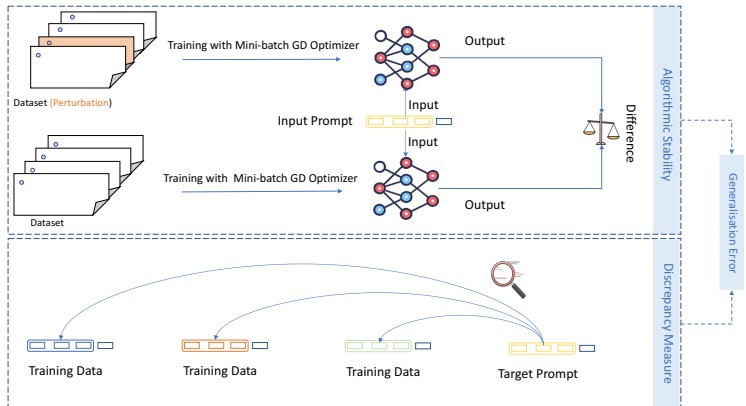

Figure 1: Algorithmic stability quantifies the sensitivity of an algorithm to perturbations in the training data, where higher stability (i.e., lower sensitivity) typically indicates better generalisation. Discrepancy measures the divergence between the target distribution and the training data distribution, assessing how well the training data represents the target data.

Our goal is to predict unknown sequence $\mathbf{C}^{N+1}$, based on the given ICL prompt $\mathbf{P}^i$. The corresponding population version is

$$\mathcal{L}(\mathcal{T}) = \frac{1}{N_c} \sum_{j=1}^{N_c} \mathbb{E}\left[\ell(\mathcal{T}(\mathbf{P}^{N+1,j-1})_{*,:}, C_j^{N+1})\right]. \tag{1}$$

Additionally, the expected risk, which takes error accumulation into account, is expressed as

$$\mathcal{L}^{EA}(\mathcal{T}) = \frac{1}{N_c} \sum_{j=1}^{N_c} \mathbb{E}\left[\ell(\mathcal{T}(\hat{\mathbf{P}}^{N+1,j-1})_{*,:}, C_j^{N+1})\right], \tag{2}$$

For notational simplicity, we use $\theta = \{O_m^l, V_m^l, Q_m^l, K_m^l, W_1^l, W_2^l\}_{l=1,m=1}^{L,N_a}$, to represent all trainable parameters. Moreover, the training details using mini-batch GD is summarized in Algorithm 1.

## 4 THEORETICAL ANALYSIS

We assume that $\mathcal{T}_S(\cdot)$ denotes a Transformer model trained using mini-batch GD on the dataset $S = \{(\mathbf{p}^i, \mathbf{c}^i)\}_{i=1}^N$. The main concern is how to bound the generalisation error in terms of the difference between the population risk and the empirical risk evaluated at $\mathcal{T}_S$. We address this question using tools from algorithmic stability and distributional discrepancy, as illustrated in Figure 1.

### 4.1 MINI-BATCH GD-DEPENDENT ALGORITHMIC STABILITY

This paper utilizes a variant of commonly-used uniform stability in statistical learning theory (Bousquet & Elisseeff, 2002). Let $S^i$ denote the dataset $S$ with its $i$-th sample replaced by an independent sample drawn from the same distribution. The algorithmic stability is defined as below.

**Definition 3.** *A randomized algorithm $\mathcal{G}$ that maps $N$-size dataset $S$ to estimator $\mathcal{T}_S$ has uniform stability $\beta$ if the following inequality holds*

$$\frac{1}{N_c} \sum_{j=1}^{N_c} \mathbb{E}_{\mathcal{G}}\left|\ell(\mathcal{T}_S(\mathbf{P}^{k,j-1})_{*,:}, C_j^k) - \ell(\mathcal{T}_{S^i}(\mathbf{P}^{k,j-1})_{*,:}, C_j^k)\right| \leq \beta, \ \forall i,k = 1,...,N, \ \forall S, S^i.$$

To establish the upper bound on uniform algorithmic stability, we introduce the following assumptions.

**Assumption 1.** *[Boundedness] The norm of each row of the input prompt $\mathbf{P}^i$ and the norm of each response vector $C_j^i$, for $j = 1, \ldots, N_c$, $i = 1, \ldots, N$, are uniformly bounded by constants $B_P$ and*

$B_C$, respectively. Additionally, for any attention head $m = 1, \ldots, N_a$ and any layer $l = 1, \ldots, L$, the parameter norms satisfy the following conditions $\|W_1^l\|_2 \leq B_{W_1}$, $\|W_2^l\|_2 \leq B_{W_2}$, $\|Q_m^l\|_2 \leq B_Q$, $\|K_m^l\|_2 \leq B_Q$, $\|V_m^l\|_2 \leq B_V$, $\|O_m^l\|_2 \leq B_O$.

This mild boundedness assumption is widely utilized in various theoretical studies (Bai et al., 2024; Zhang et al., 2023). Indeed, the boundedness assumptions in our theoretical analysis can be further relaxed to unbounded settings, with the theoretical results still holding. For example, one can replace the assumption of a hard bound on inputs with a light-tailed distribution assumption (e.g., inputs or features have sub-Gaussian tails) Attia & Koren (2024). This means extremely large input values are exponentially unlikely, effectively limiting the influence of outliers without requiring an absolute bound. Under this assumption, we thus denote the maximum value of the loss function as $M_\ell = \sup \ell(\cdot)$. Additionally, to establish the bound on algorithmic stability, we consider its Lipschitz constant with respect to trainable parameters (Definition 4) and the Lipschitz smoothness constant $\gamma$ (Definition 5). Detailed calculations for both are provided in Appendix H.

**Definition 4.** *(Lipschitz constant) For a Lipschitz function $f$ defined over domain $\mathcal{X}$, the Lipschitz constant $L_f$ is defined as the smallest value such that $\|f(y) - f(x)\|_2 \leq L_f \|y - x\|_2, \forall x, y \in \mathcal{X}$.*

**Definition 5.** *(Lipschitz smooth constant) A function $f$ defined over domain $\mathcal{X}$ is said to be Lipschitz smooth if there exists a constant $\gamma > 0$ such that $\|\nabla f(x) - \nabla f(y)\|_2 \leq \gamma \|x - y\|_2$ for all $x, y \in \mathcal{X}$.*

We then give the bound on the algorithmic stability (See Appendix C for the detailed proof).

**Theorem 1.** *Let Assumption 1 be true and the learning rate be $\eta_k = \frac{1}{k^\alpha}, \alpha > 0$. The algorithmic stability satisfies*

$$
\beta \lesssim \begin{cases} \frac{BM_\ell L_\ell^{\frac{2}{\alpha(1+\gamma)}} Q^{\frac{\gamma}{1+\gamma}}}{N\gamma\alpha}, & \text{if } \gamma \leq \frac{1+\sqrt{1-4\alpha(1-\alpha)}}{2\alpha}, \\ \frac{BM_\ell L_\ell^{\frac{2}{\alpha(1+\gamma)}} Q^{\frac{\alpha\gamma^2+1-\alpha}{1+\gamma}}}{N\gamma\alpha}, & \text{if } \gamma > \frac{1+\sqrt{1-4\alpha(1-\alpha)}}{2\alpha}, \end{cases} \tag{3}
$$

where $M_\ell$, $L_\ell$, and $\gamma$ for Transformer are given in Equations (7)-(9). There constants are related to Transformer architecture, e.g., depth $L$ and the number of attention head. For example, since quantities such as $M_\ell$ grow exponentially in $L$ (see Equation (7)), a sufficient condition for stability is that the depth grows at most logarithmically with $N$.

**Remark 1.** *The algorithmic stability bound depends on the Lipschitz smoothness constant $\gamma$, batch size $B$, number of iterations $Q$, dataset size $N$, and learning rate decay $\alpha$. For small $\gamma$, stability is better controlled, while for large $\gamma$, stability degrades rapidly with $Q$, especially when $\alpha$ is small. A larger dataset $N$ improves stability, but increasing $B$ or the maximum loss $M_\ell$ worsens it. This aligns with existing studies indicating that small-batch SGD tends to yield superior generalisation performance compared to large-batch SGD or full-batch GD (Keskar et al., 2017; Masters & Luschi, 2018; LeCun et al., 2012; Wilson & Martinez, 2003). To maintain stability, it is beneficial to use smaller batch sizes, moderate $\alpha$, and smooth the loss function to keep $\gamma$ small.*

The following corollaries examine its asymptotic behavior under two distinct scenarios, characterized by the smoothness of the loss landscape.

**Corollary 1.** *[Well-conditioned Smoothness] Let the conditions in Theorem 1 be true, and $\zeta_1$ and $\zeta_2$ be arbitrary non-negative real numbers that control the growth rates of the batch size and iteration count, respectively. If the loss landscape is sufficiently smooth, i.e., $\gamma \leq (2\alpha)^{-1}(1+\sqrt{1-4\alpha(1-\alpha)})$, and the upper bound $M_\ell$, Lipschitz (smooth) constants $L_\ell$ and $\gamma$ are bounded. By putting $|B| = O(N^{\zeta_1})$ and $Q = O(N^{\zeta_2})$ into Eq. (3), the upper bound on algorithmic stability is $\beta = O(N^{\zeta_1 + \frac{\zeta_2\gamma}{1+\gamma} - 1})$.*

Some techniques such as regularization can be used to ensure that the loss landscape is smooth. Corollary 1 captures a fundamental trade-off between optimization and stability. Increasing the number of iterations $Q$ (and/or the batch size) generally improves optimization and reduces the empirical risk (which is observable). At the same time, our stability analysis shows that larger $Q$ amplifies the accumulated perturbations along the optimization path, thereby worsening the stability coefficient $\beta$ and enlarging the generalization gap.

**Corollary 2.** *[Insufficient Smoothness] Let the conditions in Theorem 1 be true. If $\gamma > (2\alpha)^{-1}(1 + \sqrt{1-4\alpha(1-\alpha)})$, by putting $|B| = O(N^{\zeta_1})$, $Q = O(\ln N)$ into Eq. (3), we get $\beta = \tilde{O}(N^{\zeta_1 - 1})$.*

Corollary 2 indicates that when the Lipschitz smoothness constant is overly large, constraining iteration growth to a logarithmic scale effectively mitigates instability.

## 4.2 Discrepancy Measure

Given the potential distribution shift between training and target data, a suitable metric that does not impose distributional assumptions is essential for quantifying their divergence. This paper extends a discrepancy metric inspired by Kuznetsov & Mohri (2015) to make it hypothesis-space independent.

**Definition 6.** *(Discrepancy Measure) For the estimator $\mathcal{T}_S$, the discrepancy measure is defined as*

$$\text{disc}(\mathbf{q}) := \frac{1}{N_c} \sum_{j=1}^{N_c} \Big[ E_{N+1,j} - \sum_{i=1}^{N} q_i E_{i,j} \Big],$$

*where $E_{i,j} = \mathbb{E}\big[\ell(\mathcal{T}_S(\mathbf{P}^{i,j-1})_{*,:}, C_j^i)|\{(\mathbf{p}^m, \mathbf{c}^m)\}_{m=1}^{i-1}\big]$.*

The $\text{disc}(\mathbf{q})$ measures the degree of misalignment between the target task distribution and the training distribution. We then show how this discrepancy can be quantified under different scenarios.

**I.i.d. Scenario**: In the ideal i.i.d. case, where the training and target distributions match, the discrepancy admits the following asymptotic property (see Appendix E for proof).

**Theorem 2.** *Let $\mathcal{T}_S$ be a learning algorithm that is uniformly $\beta$-stable. Suppose the training data and test sample are i.i.d.. Then, with confidence at least $1 - \delta, \forall \delta \in (0, 1)$, the discrepancy satisfies $\text{disc}(\mathbf{q}) \leq 2\beta\|\mathbf{q}\|_2 N\sqrt{\log(2/\delta)}$, where $\beta$ is defined in Eq equation 3, and thus $\text{disc}(\mathbf{q}) \to 0$ as $N \to \infty$ provided that $\beta\|\mathbf{q}\|_2 N \to 0$.*

The condition $\beta\|\mathbf{q}\|_2 N \to 0$ is easy to satisfy under standard choices of the training weights. For example, if we take uniform weights $q_i = 1/N$ for all samples, then $\|\mathbf{q}\|N = N^{1/2}$, and thus the requirement becomes simply $\beta = o(N^{-1/2})$. Theorem 1 shows that such a decay rate for $\beta$ is achievable under multiple concrete regimes. For instance, Corollary 1 implies that $\beta = o(N^{-1/2})$ holds whenever $\zeta_1 + \zeta_2\gamma/(1 + \gamma) < 1/2$, where $\zeta_1$ and $\zeta_2$ characterize the growth rates of the batch size and the iteration count $Q$, and $\gamma$ is the Lipschitz-smoothness parameter of the loss.

**Non-i.i.d Scenario**: If the target domain is entirely unrelated to the training domains, achieving accurate predictions becomes nearly impossible. Therefore, we consider a scenario where at least some training domains share a meaningful relationship with the target domain. Formally, suppose that there exists an effective prompt such that the example distribution set is drawn from a distribution related to the training distributions, ensuring $\frac{1}{N_c} \sum_{j=1}^{N_c} \big[E_{N+1,j} - \sum_{i \in \mathcal{I}} v_i E_{i,j}\big] \leq \epsilon, \epsilon > 0$, where $\mathcal{I} \subset \{1, ..., N\}$ is the index set that refers to the related training data, and $v_i$ is the corresponding weight. Techniques such as incorporating more diverse training data and designing more effective ICL prompts can help reduce $\epsilon$ by better aligning the training and test environments. For this non-i.i.d scenario, Theorem F provides an sequential Rademacher complexity based upper bound on $\text{disc}(\mathbf{q})$ (See Appendix F for detailed proof).

**Theorem 3.** *Under the above situation and Assumption 1, with confidence at least $1 - \delta$, there holds*

$$\begin{aligned}
\text{disc}(\mathbf{q}) &\leq \epsilon + \sup_{\mathcal{T} \in \mathcal{H}} \left\{ \frac{1}{N_c} \sum_{i=1}^{N} \sum_{j=1}^{N_c} (v_i - q_i)\ell(\mathcal{T}(\mathbf{p}^{i,j})_{*,:}, C_j^i) \right\} + 3M_\ell\sqrt{\pi \log N}\mathcal{R}_N(\{\ell \circ \mathcal{T}\}) \\
&+ M_\ell\|\mathbf{q} - \mathbf{v}\|_2\sqrt{2 \log \frac{1}{\delta}}
\end{aligned}$$

*where the sequential Rademacher complexity $\mathcal{R}_N(\{\ell \circ \mathcal{T}\})$ over measurable hypothesis space $\mathcal{H}$ (see Definition 7 for more details) satisfies $\mathcal{R}_N(\{\ell \circ \mathcal{T}\}) = 4RL_{\mathcal{T}}^*\sqrt{N_p + N_c}B_P\|\mathbf{q} - \mathbf{v}\|$, $L_{\mathcal{T}}^*$ is the Lipschitz constant given in Eq. (12), and $R = \max\left\{B_C, (B_{W_1}B_{W_2}B_V B_O N_a)^L B_P\right\}$.*

**Remark 2.** *In the non-i.i.d. setting, Theorem 3 reveals how the complexity of the hypothesis space involved in the second and third terms affects disc($\mathbf{q}$). For instance, a more complex hypothesis space, characterized by higher sequential Rademacher complexity, allows the model to fit arbitrary*

*patterns in the training prompts, increasing its sensitivity to distribution shift and thereby amplifying the discrepancy. It suggests that regularization techniques, such as weight norm constraints, may help control this complexity and thus improve alignment between training and testing distributions. In addition, the weight discrepancy $\|\mathbf{q} - \mathbf{v}\|$ offers a theoretical explanation for the effectiveness of finetuning, which reweights training samples toward those relevant to the target.*

### 4.3 GENERALISATION ERROR ANALYSIS

Building on the above analysis, this section derives an upper bound on the generalisation errors $\mathcal{L}(\mathcal{T}_S) - \hat{\mathcal{L}}(\mathcal{T}_S)$. The detailed proof is provided in Appendix D.

**Theorem 4.** *Under Assumption 1, let $\mathcal{T}_S$ be a $\beta$-stable learning algorithm and $\mathbf{q} = (q_1, \cdots, q_{N_c})$ be any weight vector used in training objective. For any $\delta > 0$, each of the following bounds holds with confidence at least $1 - \delta$:*

$$\mathcal{L}(\mathcal{T}_S) \quad \leq \quad \frac{1}{N_c} \sum_{i=1}^{N} \sum_{j=1}^{N_c} q_i \ell(\mathcal{T}_S(\mathbf{p}^{i,j})_{*,:}, c_j^i) + \text{disc}(\mathbf{q}) + \|\mathbf{q}\|_1 \beta + 2\|\mathbf{q}\|_2 M_\ell \sqrt{2 \log \frac{4}{\delta}},$$

*where $\beta$ is defined in Eq. (3).*

The following corollaries provide a more detailed characterization of the asymptotic behavior under both i.i.d. and non-i.i.d. data settings.

**Corollary 3** (Asymptotic Behavior under i.i.d Scenarios). *Let the conditions in Theorem 4 and i.i.d. assumption be true. Let $q_i = \frac{1}{N}$, $|B| = O(N^{\zeta_1})$, and $Q = O(N^{\zeta_2})$. a) When the loss function scape is well-conditioned smoothness, with confidence at least $1 - \delta$, $0 < \delta < 1$, there holds*

$$\mathcal{L}(\mathcal{T}_S) \lesssim \frac{1}{N_c N} \sum_{i=1}^{N} \sum_{j=1}^{N_c} \ell(\mathcal{T}_S(\mathbf{p}^{i,j})_{*,:}, c_j^i) + N^{-\frac{1}{2}} \sqrt{2 \log \frac{4}{\delta}}$$

*when $\zeta_1 + \frac{\zeta_2 \gamma}{1+\gamma} = \frac{1}{2}$. b) When Lipschitz smoothness constant is large such that $\gamma > (2\alpha)^{-1}(1 + \sqrt{1 - 4\alpha(1 - \alpha)})$, by setting $|B| = O(N^{\zeta_1})$, $\zeta_1 \leq \frac{1}{2}$, and $Q = O(\ln N)$, there holds*

$$\mathcal{L}(\mathcal{T}_S) \lesssim \frac{1}{N_c N} \sum_{i=1}^{N} \sum_{j=1}^{N_c} \ell(\mathcal{T}_S(\mathbf{p}^{i,j})_{*,:}, c_j^i) + N^{-\frac{1}{2}} \sqrt{2 \log \frac{4}{\delta}}$$

*with at least confidence $1 - \delta$.*

**Remark 3.** *In the ideal i.i.d. setting, the corollary above establishes how the generalisation error bound achieves the fastest convergence rate of $O(N^{-\frac{1}{2}})$ under different levels of loss landscape smoothness. Specifically, when the loss function is sufficiently smooth, the hyper-parameters $\zeta_1, \zeta_2$ are tuned such that $2\zeta_1 + \frac{2\zeta_2 \gamma}{1+\gamma} = 1$. However, when the smoothness constant is large, exceeding the threshold $(2\alpha)^{-1}(1 + \sqrt{1 - 4\alpha(1 - \alpha)})$, to achieve the convergence rate $O(N^{-\frac{1}{2}})$, the number of iterations is recommended to scale logarithmically with the sample size.*

| Scenario | Parameter Settings | Convergence Rate |
|---|---|---|
| I.i.d & Smooth | $\|B\| = O(N^{\zeta_1}), Q = O(N^{\zeta_2}), 2\zeta_1 + \frac{2\zeta_2\gamma}{1+\gamma} = 1$ | $O(N^{-\frac{1}{2}})$ |
| I.i.d & Non-smooth | $\|B\| = O(N^{\zeta_1}), Q = O(\ln N), \zeta_1 \leq \frac{1}{2}$ | $O(N^{-\frac{1}{2}})$ |
| Non-i.i.d & Smooth | $\|\mathbf{q} - \mathbf{v}\| + \|\mathbf{q}\| = O(N^{\zeta_3})$, $\|B\| = O(N^{\zeta_1}), Q = O(N^{\zeta_2}), \zeta_1 + \frac{\zeta_2\gamma}{1+\gamma} < 1$ | $O(N^{\max\{\zeta_3, \zeta_1 + \frac{\zeta_2\gamma}{1+\gamma} - 1\}})$ |
| Non-i.i.d & Non-smooth | $\|\mathbf{q} - \mathbf{v}\| = O(N^{\zeta_3}), \|\mathbf{q}\| = O(N^{\zeta_4})$, $\|B\| = O(N^{\zeta_1}), Q = O(\ln N), N_p = O(N^{\zeta_2})$ | $O(N^{\max\{2L\zeta_2 + \zeta_3, \zeta_4, \zeta_1 - 1\}})$ |

Table 2: Summary of Generalisation Error Bounds under Different Scenarios.

**Corollary 4.** *[Asymptotic Behavior under Non-i.i.d Scenarios] Let the conditions in Theorem 4 and 3 be true. a) If the loss landscape is sufficiently smooth and if $\|\mathbf{q} - \mathbf{v}\| + \|\mathbf{q}\| = O(N^{\eta_3})$, then by setting $|B| = O(N^{\zeta_1})$ and $Q = O(N^{\zeta_2})$, for any $\delta > 0$, with confidence at least $1 - \delta$, there holds:*

$$\mathcal{L}(\mathcal{T}_S) \quad \leq \quad \frac{1}{N_c} \sum_{i=1}^{N} \sum_{j=1}^{N_c} q_i \ell(\mathcal{T}_S(\mathbf{p}^{i,j})_{*,:}, c_j^i) + \sup_{\mathcal{T} \in \mathcal{H}} \left\{ \sum_{i=1}^{N} (v_i - q_i) \ell(\mathcal{T}(\mathbf{p}^{i,j})_{*,:}, c_j^i) \right\}$$

$$+ \quad N^{\max\{\zeta_1 + \frac{\zeta_2\gamma}{1+\gamma} - 1, \zeta_3\}} \sqrt{2 \log \frac{4}{\delta}} + \epsilon.$$

*b) If* $\|\mathbf{q} - \mathbf{v}\| = O(N^{\eta_3})$ *and* $\|\mathbf{q}\| = O(N^{\eta_4})$, *then by setting* $|B| = O(N^{\zeta_1})$, $Q = O(\ln N)$, *and the ICL prompt length as* $N_p = O(N^{\zeta_2})$, *for any* $\delta > 0$, *with probability at least* $1 - \delta$, *there holds*

$$\mathcal{L}(\mathcal{T}_S) \;\leq\; \frac{1}{N_c} \sum_{i=1}^{N} \sum_{j=1}^{N_c} q_i \ell(\mathcal{T}_S(\mathbf{p}^{i,j})_{*,:}, c_j^i) + \sup_{\mathcal{T} \in \mathcal{H}} \left\{ \sum_{i=1}^{N} (v_i - q_i) \ell(\mathcal{T}(\mathbf{p}^{i,j})_{*,:}, c_j^i) \right\}$$

$$+ \; N^{\max\{2L\zeta_2 + \zeta_3, \zeta_1 - 1, \zeta_4\}} \sqrt{2 \log \frac{4}{\delta}} + \epsilon.$$

**Remark 4.** *Corollary 4 characterizes ICL generalization under non-i.i.d. settings by establishing two upper bounds under distinct smoothness conditions. The results show that smoother loss landscapes and better alignment between training and test prompt distributions (i.e., small* $\|\mathbf{q} - \mathbf{v}\|$*) yield improved generalization.*

**Remark 5.** *From Corollary 4, to achieve better cross-domain generalization (i.e., minimizing* $\mathcal{L}(\mathcal{T}_S)$*), we shall minimize the following optimization problem:*

$$\min_{\mathbf{q}} \left\{ \frac{1}{N_c} \sum_{i=1}^{N} \sum_{j=1}^{N_c} q_i \ell(\mathcal{T}_S(\mathbf{p}^{i,j})_{*,:}, c_j^i) + \sup_{\mathcal{T} \in \mathcal{H}} \left\{ \sum_{i=1}^{N} (v_i - q_i) \ell(\mathcal{T}(\mathbf{p}^{i,j})_{*,:}, c_j^i) \right\} + \lambda_1 \|\mathbf{s} - \mathbf{q}\|_2^2 + \lambda_2 \|\mathbf{q}\|_2^2 \right\},$$

(4)

*where* $\lambda_1$ *and* $\lambda_2$ *are regularization parameters. The entire optimization procedure can be decomposed into two stages. In the first stage, we solve for the optimal sample-weight vector* $\mathbf{q}$ *by optimizing the latter three terms. This subproblem can be computed via DC programming (Tao & An, 1998) or gradient-based methods. Once the optimal sample weights have been obtained, we then optimize the first term accordingly to learn the final model parameters.*

In practical scenarios, the model typically uses its own estimated token to predict subsequent tokens. This approach, by its nature, leads to cumulative errors as inaccuracies in earlier steps propagate forward. The corresponding theoretical result and proof are provided in Appendix G.

## 5 NUMERICAL EVALUATION

Our experimental setup follows (Li et al., 2023), where all evaluations are conducted using the same GPT-2 architecture implemented via the HuggingFace Transformers library (Wolf et al., 2020), consisting of 12 layers and 8 attention heads. All empirical evaluations are conducted using NVIDIA H20 GPUs with 80GB of memory.

**Evaluation on i.i.d data scenario:** In the ideal i.i.d. setting, we focus on validating the asymptotic behavior predicted by Corollary 3 and the error accumulation characterized in Theorem 5. We consider a $d = 10$-dimensional linear regression task, where each in-context example is of the form $(\mathbf{p}, \mathbf{c})$. For each sample $i$, given a parameter vector $\beta^i \in \mathbb{R}^d$, we generate a length-$L$ sequence using the recurrence relation $c_l^i = \beta_{l-1}^i c_{l-1}^i + \epsilon$, for $l = 1, \ldots, L$, where the initial query $c_0^i \sim \mathcal{N}(0, 0.1\mathbf{I}_d)$, and the noise term $\epsilon \sim \mathcal{N}(0, 0.1\mathbf{I}_d)$. The prompt $\mathbf{p}$ is constructed by concatenating two such examples along with the query input $c_0^i$ into a single flattened vector. Each parameter vector $\beta^i \in \mathbb{R}^d$ is independently sampled from $\mathcal{N}(0.1, 0.1\mathbf{I}_d)$. We set the sample size $N \in \{50, 100, 200, 400, 800, 1600\}$, use uniform training weights $q_i = 1/N$, and set the batch size to $|B| = N^{1/2}$ to ensure sufficient training. For evaluation, we independently generate 1000 i.i.d. test samples. We fix the number of optimization iterations to $Q = 200$, set the learning rate decay exponent to $\alpha = 1$, and systematically vary the sequence length $N_c \in \{1, 2, 3, 4, 5, 6, 7, 8, 9\}$.

*The Generalisation Error Convergence Analysis:* We evaluate the generalization error as the number of training samples $N$ increases. Figure 2(a) demonstrates that the error decreases and asymptotically vanishes as $N \to \infty$, consistent with the theoretical prediction in Corollary 3 for the i.i.d. setting. Results are shown for sequence lengths 1 and 2; other lengths follow similar trends but are omitted due to large differences in error magnitude, which would obscure the overall pattern if plotted together.

*The Error Accumulation Analysis:* Figure 2(b) shows that the generalization error increases with sequence length, following an approximately polynomial trend. In particular, once the sequence length exceeds a threshold near $\ln N$, the error rises sharply. Moreover, this threshold shifts to larger values as the sample size increases. These empirical findings support Theorem 5.

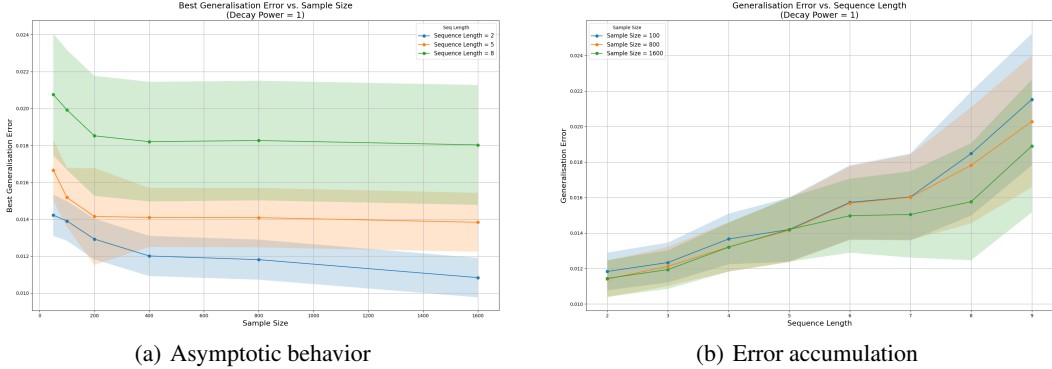

(a) Asymptotic behavior          (b) Error accumulation

Figure 2: The generalisation error under i.i.d scenario. For extended results under non-i.i.d. distributions, additional sequence lengths and overfitting risk, please refer to Figures 4 and 5 in Appendix I.

In addition to the i.i.d. scenario presented above, we also conduct evaluations under non-i.i.d. settings, which are detailed in Appendix I. These experiments are designed to assess the robustness of our theoretical claims, particularly under distribution shift conditions where training and test domains exhibit structural divergence. The results demonstrate consistent alignment with our theoretical bounds, especially regarding the influence of distributional discrepancy and prompt reweighting.

## 6 CONCLUSION

This study derives ICL generalisation error bound with asymptotic convergence analysis by examining algorithmic stability under mini-batch GD and a distribution-level discrepancy measure. Our results reveal how optimization settings interact with the smoothness of the loss landscape to ensure algorithmic stability, and how, when combined with high-quality prompts, they enable effective ICL generalization. On the theoretical side, future work should develop tighter generalization bounds using techniques such as gradient stability. On the practical side, our findings inform algorithm design, including strategies like weighted training samples.

## ETHICS STATEMENT

This paper is theoretical and does not involve human subjects, personally identifiable information, or sensitive data. All proofs and theoretical analyses are conducted under standard mathematical assumptions and are intended to advance the understanding of large models' generalization ability. We foresee no ethical concerns with the content or potential applications of this work.

## REPRODUCIBILITY STATEMENT

Our work is primarily theoretical, and all theoretical results are presented with formal statements, clearly defined assumptions, and complete proofs provided in Section 5 and Appendix I. To support the practical relevance of our findings, we include empirical experiments conducted using publicly available open-source models, specifically, the GPT-2 architecture implemented via the HuggingFace Transformers library (Wolf et al., 2020). All implementation details, including model configurations and data processing steps, are described in the Numerical Evaluations section. No proprietary data or code is used, and we are committed to making our results fully reproducible.

## ACKNOWLEDGMENTS

This project is supported by the National Research Foundation, Singapore, under its NRF Professorship Award No. NRF-P2024-001.

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

# Appendix

## CONTENTS

## A    NOTATIONS

For clarity and ease of reference, Table 3 presents a comprehensive summary of the notations used throughout this paper. The input and ICL variable spaces for the $i$-th sample are denoted by $\mathcal{X}^i$ and $\mathcal{C}^i$, respectively, while $X^i$ and $C^i$ represent the corresponding input and ICL random variables. Their specific realizations are given by $x^i$ and $c^i$. The dataset consists of $N$ samples, with $N_p$ denoting the length of the prompt $\mathbf{P}$, $N_e$ the number of demonstration examples, and $N_c$ the number of steps in the ICL inference process. The Transformer model employs $N_a$ self-attention heads, and the batch size in the mini-batch GD optimization scheme is denoted as $|B|$. The ICL prompt for the $i$-th sample, containing $N_e$ examples followed by a query, is represented by $\mathbf{P}^i$, while $\mathbf{P}^{i,j}$ extends this by incorporating $j$ additional reasoning steps. The estimated version of this prompt is given by $\hat{\mathbf{P}}^{i,j}$. The parameters associated with the $m$-th attention module in the $l$-th layer of the Transformer are represented as $O_m^l$, $V_m^l$, $Q_m^l$, and $K_m^l$, corresponding to the output, value, query, and key matrices, respectively, while $W_1^l$ and $W_2^l$ denote the parameters of the MLP in the $l$-th layer. Finally, the empirical risk is denoted by $\hat{\mathcal{L}}(\theta)$, while $\mathcal{L}(\theta)$ represents the expected risk associated with the ICL prompt $\mathbf{P}$, and $\mathcal{L}^{EA}(\theta)$ denotes the expected risk when using the estimated ICL prompt $\hat{\mathbf{P}}$, accounting for potential deviations due to reasoning inaccuracies.

Table 3: Notations

| Notations | Descriptions |
|---|---|
| $\mathcal{X}^i, \mathcal{C}^i$ | the input and output variable space for $i$-th sample, respectively |
| $X^i, C^i$ | the input and output random variables for $i$-th sample, respectively |
| $x^i, c^i$ | the realizations of $X$ and $\mathbf{C}$ for $i$-th sample, respectively |
| $N$ | the sample size |
| $N_p$ | the length-$N_p$ prompt $\mathbf{P}$ |
| $N_e$ | the size of demonstrations |
| $N_c$ | the length of inference |
| $N_a$ | the number of self-attention heads |
| $|B|$ | the batch size in Mini-Batch GD optimization scheme |
| $\mathbf{P}^i$ | the $i$-th ICL prompt variable with $N_e$ examples followed by a query |
| $\mathbf{P}^{i,j}$ | the $i$-th ICL prompt variable with $N_e$ examples followed by a query and $j$ tokens |
| $\hat{\mathbf{P}}^{i,j}$ | the $i$-th ICL prompt variable with $N_e$ examples followed by a query and $j$ estimated tokens |
| $O_m^l$ | represents the parameter associated with the $m$-th attention module in the $l$-th layer |
| $V_m^l$ | represents the parameter associated with the $m$-th attention module in the $l$-th layer |
| $Q_m^l$ | represents the parameter associated with the $m$-th attention module in the $l$-th layer |
| $K_m^l$ | represents the parameter associated with the $m$-th attention module in the $l$-th layer |
| $W_1^l$ | represents the parameter associated with MLP in the $l$-th layer |
| $W_2^l$ | represents the parameter associated with MLP in the $l$-th layer |
| $\ell(\cdot)$ | the loss function |
| $\hat{\mathcal{L}}(\theta)$ | the empirical risk |
| $\mathcal{L}(\theta)$ | the expected risk associated with $\mathbf{P}$ |
| $\mathcal{L}^{EA}(\theta)$ | the expected risk associated with prompt $\hat{\mathbf{P}}$ |

## B    PROOF SKETCH

Our main results follow a structured sequence of steps. Below we summarize the logical chain of the analysis and involved technical tools.

**Step 1: Formalizing stability and discrepancy.** We begin by defining two key notions: (i) the *algorithmic stability* $\beta$, which measures how sensitive the mini-batch SGD-trained Transformer is to replacing a single training example; and (ii) the *discrepancy measure* disc($\mathbf{q}$), which quantifies the distributional mismatch between the importance-weighted training distribution and the target prompt distribution. These quantities jointly determine the generalization behavior of in-context prediction.

**Step 2: Decomposing generalisation error.** We establish the general decomposition (see the proof of Theorem 4 in Section D):

$$\mathcal{L}(\mathcal{T}_S) \leq \hat{\mathcal{L}}(\mathcal{T}_S) + \text{disc}(\mathbf{q}) + \beta + \text{(vanishing statistical term)}, \tag{5}$$

which shows that the generalization error consists of the training loss term, the distribution-shift term $\mathrm{disc}(\mathbf{q})$, the algorithmic stability $\beta$ and vanishing statistical term. Hence, to obtain explicit bounds on $\mathcal{L}(\mathcal{T}_S)$, we must control both $\beta$ and $\mathrm{disc}(\mathbf{q})$.

**Step 3: Bounding the stability of Transformers under mini-batch SGD.** To control $\beta$, we analyze how perturbing a single training sample influences the multi-head, multi-layer Transformer during $Q$ mini-batch SGD updates. Using the layer-wise Lipschitz and smoothness constants derived in Appendix H, we derive a recurrence relating the perturbed and unperturbed updates. Solving this recurrence yields the stability bounds in Theorem 1.

**Step 4: Bounding discrepancy.** We next characterize $\mathrm{disc}(\mathbf{q})$ under both i.i.d (Theorem 2) and non-i.i.d scenarios (Theorem 3) by employing concentration inequalities together with the notion of Sequential Rademacher Complexity (see Definition 7 in Appendix E-F).

**Step 5: Combining the bounds.** Substituting the stability bound (Step 3) and discrepancy bounds (Step 4) into the decomposition (Step 2) yields our final generalization results.

Finally, Figure 3 (see Appendix B) outlines the technical tool used for our theoretical analysis.

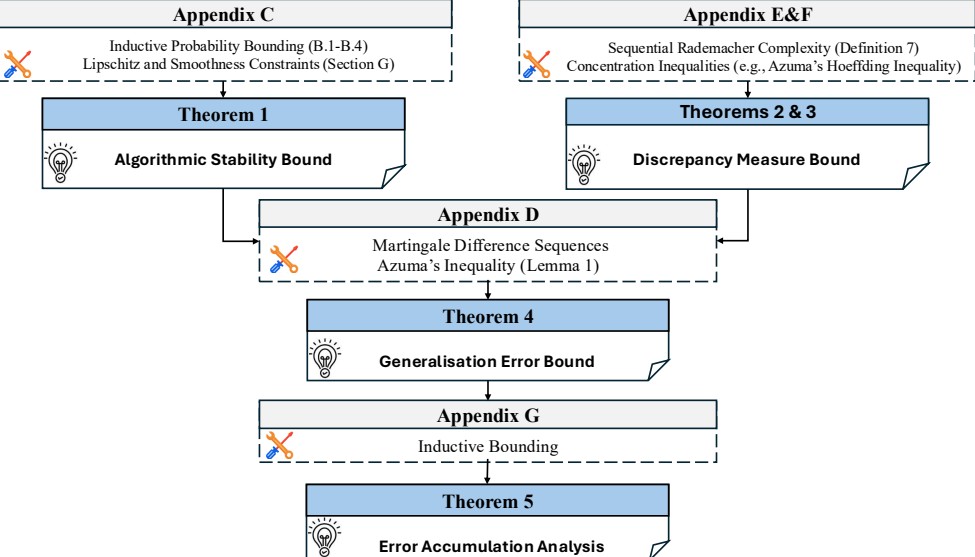

Figure 3: Proof sketch: the logical dependencies among stability, discrepancy, generalisation, and error accumulation analyses.

## C  ALGORITHMIC STABILITY (PROOF OF THEOREM 1)

Building on the insights from the algorithmic stability bound for SGD under bilevel optimization (Bao et al., 2021), this section derives an upper bound on algorithmic stability of the Transformer model when trained with the mini-batch GD optimizer.

For any sample pairs $(\mathbf{P}^{i,j}, C_j^i)$, we give a general version of expected risk with weights $v_j, j = 1, ..., N_c$, which is defined by

$$\mathcal{L}(\mathcal{T}) = \sum_{j=1}^{N_c} v_j \mathbb{E}\left[\ell(\mathcal{T}(\mathbf{P}^{i,j-1}), C_j^i)\right]. \tag{6}$$

Let $\mathcal{T}_S^{(q)}$ represent the optimization process after $q$ iterations, following Algorithm 1. Define $\delta_q = \|\theta^{(q)} - \theta'^{(q)}\|_F$, where $\theta^{(q)}$ and $\theta'^{(q)}$ are the respective outputs of $\mathcal{T}_S^{(q)}$ and $\mathcal{T}_{S'}^{(q)}$, where $S$ and $S'$ differ by a single data point.

Under Assumption 1, we establish the following bound:

$$
\begin{aligned}
\mathbb{E}[|\mathcal{L}(\mathcal{T}_S^{(q)}) - \mathcal{L}(\mathcal{T}_{S'}^{(q)})|] &= \mathrm{Prob}(\delta_{q_0} = 0)\mathbb{E}\left[|\mathcal{L}(\mathcal{T}_S^{(q)}) - \mathcal{L}(\mathcal{T}_{S'}^{(q)})||\delta_{q_0} = 0\right] \\
&+ \left[\mathrm{Prob}(\delta_{q_0} > 0)\mathbb{E}[|\mathcal{L}(\mathcal{T}_S^{(q)}) - \mathcal{L}(\mathcal{T}_{S'}^{(q)})||\delta_{q_0} > 0\right] \\
&= \mathrm{Prob}(\delta_{q_0} = 0)\sum_{j=1}^{N_c} v_j \mathbb{E}\left[|\ell(\mathcal{T}_S^{(q)}(\mathbf{P}^{i,j-1}), C_j^i) - \ell(\mathcal{T}_{S'}^{(q)}(\mathbf{P}^{i,j-1}), C_j^i)||\delta_{q_0} = 0\right] \\
&+ \mathrm{Prob}(\delta_{q_0} > 0)\sum_{j=1}^{N_c} v_j \mathbb{E}\left[|\ell(\mathcal{T}_S^{(q)}(\mathbf{P}^{i,j-1}), C_j^i) - \ell(\mathcal{T}_{S'}^{(q)}(\mathbf{P}^{i,j-1}), C_j^i)||\delta_{q_0} > 0\right] \\
&\leq \sum_{j=1}^{N_c} v_j L_{j,\ell}\mathbb{E}\left[\delta_q|\delta_{q_0} = 0\right] + \mathrm{Prob}(\delta_{q_0} > 0)\sum_{j=1}^{N_c} v_j M_{j,\ell},
\end{aligned}
$$

where $L_{j,\ell}$ is the Lipschitz constant of the loss function $\ell$ with respect to $\theta$ associated with $(\mathbf{P}^{i,j}, C_j^i)$, and $M_{j,\ell}$ is the upper bound of the loss function $\ell$ associated with $(\mathbf{P}^{i,j}, C_j^i)$.

## C.1 BOUNDING PROBABILITY TERMS

If the optimization algorithm $\mathcal{T}_S^{(q_0)}$ does not select the $i$-th sample within the first $q_0$ iterations, then $\delta_{q_0} = 0$. By induction, we obtain:

$$
\mathrm{Prob}(\delta_{q_0} = 0) = \left(1 - \frac{C_{N-1}^{B-1}}{C_N^B}\right)^{q_0} = \left(1 - \frac{B}{N}\right)^{q_0} \geq 1 - \frac{Bq_0}{N}.
$$

Thus, we also have:

$$
\mathrm{Prob}(\delta_{q_0} > 0) \leq \frac{Bq_0}{N}.
$$

Correspondingly, we have $\mathrm{Prob}(\delta_{q_0} > 0) \leq \frac{Bq_0}{N}$. As a result,

$$
\mathbb{E}\left[|\mathcal{L}(\mathcal{T}_S^{(q)}) - \mathcal{L}(\mathcal{T}_{S'}^{(q)})|\right] \leq \sum_{i=1}^{N-c} v_i L_{i,\ell}\mathbb{E}\left[\delta_q|\delta_{q_0} = 0\right] + \frac{Bq_0}{N}\sum_{i=1}^{N-c} v_i M_{i,\ell}.
$$

## C.2 RECURSIVE BOUND ON $\mathbb{E}[\delta_q|\delta_{q_0} = 0]$

Let $v^{(q)} = \frac{\eta_q}{|B_q|}\sum_{i \in B_q} \nabla_\theta \hat{\mathcal{L}}(\mathcal{T})$, and let $v^{'(q)}$ be its counterpart using perturbed data. Denote $\gamma = \sum_{i=1}^{N_c} v_i \gamma_i$ by the Lipschitz smooth constant. The update rule in Algorithm 1 gives:

$$
\begin{aligned}
\mathbb{E}[\delta_q|\delta_{q_0} = 0] &= \mathrm{Prob}(1 \in B_q)\mathbb{E}[\delta_q|\delta_{q_0} = 0, 1 \in B_q] + \mathrm{Prob}(1 \notin B_q)\mathbb{E}[\delta_q|\delta_{q_0} = 0, 1 \notin B_q] \\
&= \frac{B}{N}\mathbb{E}[\|\theta^{(q-1)} - \theta^{'(q-1)} + \eta_{q-1}(v^{'(q-1)} - v^{(q-1)})\||\delta_{q_0} = 0] \\
&+ \frac{N-B}{N}\mathbb{E}\left[\|\theta^{(q-1)} - \theta^{'(q-1)} + \eta_{q-1}(v^{'(q-1)} - v^{(q-1)})\||\delta_{q_0} = 0\right] \\
&\leq C_{q-1}\mathbb{E}\left[\delta_{q-1}|\delta_{q_0} = 0\right] + D_{q-1},
\end{aligned}
$$

where

$$
C_{q-1} = \frac{B + (N-B)(1 + \eta_{q-1}\gamma)}{N}, \quad D_{q-1} = \frac{2\eta_{q-1}L_\ell B}{N}.
$$

By induction, we obtain:

$$
\mathbb{E}[\delta_q|\delta_{q_0} = 0] \leq \sum_{j=q_0}^{q-1} D_j \prod_{k=j+1}^{q-1} C_k.
$$

## C.3 BOUNDING $\prod C_k$

Since
$$C_q = 1 + (1 - B/N)\eta_q\gamma,$$
using the inequality $1 + x \le e^x$, we obtain

$$\prod_{k=j+1}^{q} C_k \le \exp\left((1 - B/N)\gamma \sum_{k=j+1}^{q} \eta_k\right).$$

Thus,

$$\mathbb{E}[\delta_q|\delta_{q_0} = 0] \le \sum_{j=q_0}^{q-1} D_j \exp\left((1 - B/N)\gamma \sum_{k=j+1}^{q-1} \eta_k\right).$$

## C.4 FINAL BOUND ON $\beta$

Combining the above results, we get

$$\mathbb{E}\left[|\mathcal{L}(\mathcal{T}_S^{(Q)}) - \mathcal{L}(\mathcal{T}_{S'}^{(Q)})|\right] \le \sum_{j=q_0+1}^{Q} D_j \exp\left(\frac{N-B}{N}\gamma \sum_{k=j+1}^{Q} \eta_k\right) \sum_{i=1}^{N_c} v_i L_{i,\ell} + \frac{Bq_0}{N}\sum_{i=1}^{N_c} v_i M_{i,\ell}.$$

Denote by $L_\ell = \sum_{i=1}^{N_c} v_i L_{i,\ell}$ and $M_\ell = \sum_{i=1}^{N_c} v_i M_{i,\ell}$, where $L_{i,\ell}$, $M_{i,\ell}$ and $\gamma_i$ are obtained in Section H.9. Finally, optimizing $q_0$ leads to the stability bound:

$$\beta \le \min_{q_0 \in \{1,\dots,Q\}} \left\{ L_\ell \sum_{j=q_0+1}^{Q} D_j \exp\left(\gamma \sum_{k=j+1}^{Q} \eta_k\right) + \frac{M_\ell B q_0}{N} \right\}.$$

We denote the original objective function by

$$
\begin{aligned}
H(q_0) &= L_\ell \sum_{j=q_0+1}^{Q} \frac{2\eta_j L_\ell B}{N} \exp\left(\gamma \sum_{k=j+1}^{Q} \eta_k\right) + \frac{M_\ell B q_0}{N} \\
&\le \frac{2L_\ell^2 B}{N} \sum_{j=q_0+1}^{Q} \frac{1}{j^\alpha}\left(\frac{Q^\alpha}{j^\alpha}\right)^\gamma + \frac{M_\ell B q_0}{N}, \quad \text{as } Q \to \infty, a \ge 1 \\
&\le \frac{2L_\ell^2 B Q^{\alpha\gamma}}{N} \sum_{j=q_0+1}^{Q} \left(\frac{1}{j^\alpha}\right)^{\gamma+1} + \frac{M_\ell B q_0}{N} \\
&\le \frac{2L_\ell^2 B Q^{\alpha\gamma}}{N} \frac{Q^{1-\alpha\gamma-\alpha} - q_0^{1-\alpha\gamma-\alpha}}{1 - \alpha\gamma - \alpha} + \frac{M_\ell B q_0}{N} \\
&= \frac{2L_\ell^2 B Q^{1-\alpha}}{N(1-\alpha\gamma-\alpha)} - \frac{2L_\ell^2 B Q^{\alpha\gamma} q_0^{1-\alpha\gamma-\alpha}}{N(1-\alpha\gamma-\alpha)} + \frac{M_\ell B q_0}{N} \\
&= \frac{2L_\ell^2 B}{N(1-\alpha\gamma-\alpha)}\left(Q^{1-\alpha} - Q^{\alpha\gamma} q_0^{1-\alpha\gamma-\alpha}\right) + \frac{M_\ell B q_0}{N}
\end{aligned}
$$

The goal is to minimize $H(q_0)$, ensuring that:

$$\beta \le \min_{1 \le q_0 < Q} \frac{2L_\ell^2 B}{N(1-\alpha\gamma-\alpha)}\left(Q^{1-\alpha} - Q^{\alpha\gamma} q_0^{1-\alpha\gamma-\alpha}\right) + \frac{M_\ell B q_0}{N}.$$

Setting $\frac{dH}{dq_0} = 0$, we obtain:

$$-\frac{2(1-\alpha\gamma-\alpha)L_\ell^2 B Q^{\alpha\gamma}}{N(1-\alpha\gamma-\alpha)} q_0^{-\alpha\gamma-\alpha} + \frac{M_\ell B}{N} = 0.$$

For an optimal selection of $q^*$, using $\eta_k = \frac{1}{k^\alpha}$, we approximate:

$$q^* = \left( \frac{2L_\ell^2 Q^{\alpha\gamma}}{M_\ell} \right)^{\frac{1}{\alpha(1+\gamma)}}$$

Finally, by setting $w_i = \frac{1}{N_c}$, the upper bound on stability is

$$
\begin{aligned}
\beta \quad \leq \quad & H(q^*) = \frac{2L_\ell^2 B}{N(1-\alpha\gamma-\alpha)} \left( Q^{1-\alpha} - Q^{\frac{\alpha\gamma^2+1-\alpha}{1+\gamma}} L_\ell^{\frac{2-2\alpha(1+\gamma)}{\alpha(1+\gamma)}} M_\ell^{\frac{\alpha(1+\gamma)-1}{\alpha(1+\gamma)}} \right) \\
+ \quad & M_\ell^{1-\frac{1}{\alpha(1+\gamma)}} B L_\ell^{\frac{2}{\alpha(1+\gamma)}} Q^{\frac{\gamma}{1+\gamma}} N^{-1} \\
\lesssim \quad & \alpha^{-1}\gamma^{-1} N^{-1} B M_\ell^{\frac{\alpha(1+\gamma)-1}{\alpha(1+\gamma)}} L_\ell^{\frac{2}{\alpha(1+\gamma)}} Q^{\frac{\max\{\gamma, \alpha\gamma^2+1-\alpha\}}{1+\gamma}}
\end{aligned}
$$

We finally derive the desirable result

$$
\beta \lesssim \begin{cases} \dfrac{B M_\ell^{\frac{\alpha(1+\gamma)-1}{\alpha(1+\gamma)}} L_\ell^{\frac{2}{\alpha(1+\gamma)}} Q^{\frac{\gamma}{1+\gamma}}}{N\gamma\alpha}, & \text{if } \gamma \leq \frac{1+\sqrt{1-4\alpha(1-\alpha)}}{2\alpha}, \alpha > 0, \\[3mm] \dfrac{B M_\ell^{\frac{\alpha(1+\gamma)-1}{\alpha(1+\gamma)}} L_\ell^{\frac{2}{\alpha(1+\gamma)}} Q^{\frac{\alpha\gamma^2+1-\alpha}{1+\gamma}}}{N\gamma\alpha}, & \text{if } \gamma > \frac{1+\sqrt{1-4\alpha(1-\alpha)}}{2\alpha}, \alpha > 0. \end{cases}
$$

## D  GENERALISATION ERROR BOUND (PROOF OF THEOREM 4)

This proof closely follows the approach of Theorem 8 in (Kuznetsov & Mohri, 2015), with the key distinction that we extend the analysis to a weighted average version. For the sake of completeness, we present it here, beginning with an essential concentration inequality.

**Lemma 1.** *(Azuma's Inequality) Suppose $\{Y_0, \cdots, Y_n\}$ is a martingale difference sequence with respect to the filtration $F_0 \subset F_1 \subset \cdots \subset F_N$. If*

$$a_t \leq \mathbb{E}[Y_{t+1}|F_t] \leq b_{t+1}, \quad \forall 0 \leq t \leq N,$$

*then the following probability bound holds:*

$$\text{Prob}\left( \sum_{i=1}^N |Y_i| \geq \epsilon \right) \leq 2\exp\left( -\frac{2\epsilon^2}{\sum_{i=1}^N (b_i - a_i)^2} \right).$$

For notational simplicity, we denote, for the $i$-th sample, the random variables as $Z^i = (\mathbf{P}^i, \mathbf{C}^i)$, $Z_j^i = (\mathbf{P}^{i,j-1}, C_j^i)$, and the sequence as $Z_j^{i:m} = (Z_j^i, \ldots, Z_j^m)$. We define $\hat{S}(i)$ as the sequence set

$$(Z^1, \ldots, Z^i, \tilde{Z}^{i+1}, \ldots, \tilde{Z}^N),$$

where $\tilde{Z}^i$ is independently drawn from the same distribution of $Z^i$. Now, consider the following quantities:

$$A_j^i = \mathbb{E}_{Z_j^{i+1:N}, \bar{Z}_j^{i+1}}\left[ \ell(\mathcal{T}_S, \bar{Z}_j^{i+1})|Z^{1:i} \right] - \mathbb{E}_{\bar{Z}^{i+1}}\left[ \ell(\mathcal{T}_{\hat{S}(i)}, \bar{Z}_j^{i+1})|Z^{1:i} \right],$$

and

$$B_j^i = \mathbb{E}_{Z_{j+1}^i}\left[ \ell(\mathcal{T}_{\hat{S}(i+1)}, Z_j^{i+1})|Z^{1:i} \right] - \ell(\mathcal{T}_S, Z_j^{i+1}),$$

where $\bar{Z}_j^{i+1} \sim \rho(\cdot|Z_j^{1:i})$ is independent of $Z_j^{i+1:N}$ and $\tilde{Z}_j^{i+1:N}$. By construction, we observe that:

$$\mathbb{E}_{Z_j^{i+1:N}, \tilde{Z}_j^{i+1:N}, \bar{Z}_j^{i+1}}[A_j^i] = 0,$$

and

$$\mathbb{E}_{Z_j^{i+1}, \tilde{Z}_j^{i+2:N}}[B_j^i] = 0.$$

These equations indicate that both sequences $A_j^i, j = 1, \ldots, N_c$ and $B_j^i, j = 1, \ldots, N_c$ form martingale difference sequences. By applying Azuma's Inequality (Lemma 1), for any $\delta > 0$, with probability at least $1 - \delta/2$, we obtain:

$$\sum_{i=0}^{N-1} q_i A_j^i \leq \|\mathbf{q}\|_2 M_{j,\ell} \sqrt{2\log\frac{4}{\delta}},$$

and

$$\sum_{i=1}^{N-1} q_i B_j^i \leq \|\mathbf{q}\|_2 M_{j,\ell} \sqrt{2 \log \frac{4}{\delta}},$$

where $M_{j,\ell}$ is the upper bound of the loss function associated with input $\mathbf{P}_i$.

Summing both inequalities, we obtain:

$$\sum_{i=1}^{N-1} q_i (A_j^i + B_j^i) \leq 2\|\mathbf{q}\|_2 M_{j,\ell} \sqrt{2 \log \frac{4}{\delta}}.$$

Next, we define the weighted sequences:

$$\bar{A}^i = \sum_{j=1}^{N_c} c_j A_j^i, \quad \bar{B}^i = \sum_{j=1}^{N_c} c_j B_j^i,$$

where $\sum_{j=1}^{N_c} c_j = 1$. Since these sequences also form martingale difference sequences, we apply the definition of uniform stability, which states that:

$$\left| \sum_{j=1}^{N_c} c_j E_j \right| \leq \beta,$$

where

$$E_j := \mathbb{E}_{Z_j^{i+1}} \left[ \ell(\mathcal{T}_{\hat{S}(i+1)}, Z_j^{i+1}) | Z^{1:i} \right] - \mathbb{E}_{Z_j^{i+1}} \left[ \ell(\mathcal{T}_{\hat{S}(i)}, Z_j^{i+1}) | Z^{1:i} \right].$$

Thus, with probability at least $1 - \delta$, we obtain:

$$\sum_{i=1}^{N} q_i (\bar{A}^i + \bar{B}^i) \leq 2\|\mathbf{q}\|_2 M_\ell \sqrt{2 \log \frac{4}{\delta}},$$

where $M_\ell = \sum_{i=1}^{N_c} w_i M_{i,\ell}$. By the definition of algorithmic stability, it follows that:

$$\sum_{i=1}^{N} q_i (\bar{A}^i + \bar{B}^i) \leq 2\|\mathbf{q}\|_2 M_\ell \sqrt{2 \log \frac{4}{\delta}}.$$

and

$$\sum_{i=1}^{N} q_i \sum_{j=1}^{N_c} c_j \mathbb{E}_{Z_j^{i+1:N}, \bar{Z}_j^i} [\ell(\mathcal{T}_S, \bar{Z}_j^i) | Z_j^{1:i}] \leq \sum_{i=1}^{N} q_i \sum_{j=1}^{N_c} c_j \ell(\mathcal{T}_S, Z_j^i) + \|\mathbf{q}\|_1 \beta + 2\|\mathbf{q}\|_2 M_\ell \sqrt{2 \log \frac{2}{\delta}}.$$

Finally, using the definition of discrepancy, we arrive at the final bound:

$$\mathcal{L}(\mathcal{T}_S) \leq \sum_{i=1}^{N} q_i \sum_{j=1}^{N_c} c_j \ell(\mathcal{T}_S, Z_j^i) + \mathrm{disc}(\mathbf{q}) + \|\mathbf{q}\|_1 \beta + 2\|\mathbf{q}\|_2 M_\ell \sqrt{2 \log \frac{2}{\delta}}.$$

By taking $w_i = \frac{1}{N_c}$ and combining the upper bound on $\beta$ (3), we obtain the desirable result.

## E   THE UPPER BOUND ON DISCREPANCY MEASURE UNDER I.I.D SCENARIO

**Lemma 2** (Asymptotic Vanishing of Discrepancy). *Let $\mathcal{T}_S$ be a learning algorithm that is uniformly $\beta$-stable. Suppose the training data and test sample are i.i.d., and $\|\mathbf{q}\|_1 = 1$. Then, the discrepancy term satisfies $|\mathrm{disc}(\mathbf{q})| \leq 2\beta \|\mathbf{q}\|_2 N \sqrt{\log(2/\delta)}$, and thus $\mathrm{disc}(\mathbf{q}) \to 0$ as $N \to \infty$ provided that $\beta_N \to 0$.*

*Proof.* Fix token index $j \in \{1, \ldots, N_c\}$ and define

$$D_j := \mathbb{E}[\ell(\mathcal{T}_S, \bar{Z}_j^{N+1}) \mid Z^{1:N}] - \sum_{i=1}^{N} q_i \mathbb{E}[\ell(\mathcal{T}_S, \bar{Z}_j^i) \mid Z^{1:i-1}].$$

We analyze each summand

$$\Delta_i := \mathbb{E}[\ell(\mathcal{T}_S, \bar{Z}_j^{N+1}) \mid Z^{1:N}] - \mathbb{E}[\ell(\mathcal{T}_S, \bar{Z}_j^i) \mid Z^{1:i-1}].$$

Introduce an intermediate model $h^{(i)} := \mathcal{T}_{S^{(i)}}$ trained on $S^{(i)} = S \setminus \{Z^i\}$. Decompose $\Delta_i$ as:

$$\Delta_i = \underbrace{\mathbb{E}[\ell(\mathcal{T}_S, \bar{Z}_j^{N+1}) \mid Z^{1:N}] - \mathbb{E}[\ell(\mathcal{T}_S, \bar{Z}_j^i) \mid Z^{1:N}]}_{(A)} + \underbrace{\mathbb{E}[\ell(\mathcal{T}_S, \bar{Z}_j^i) \mid Z^{1:N}] - \mathbb{E}[\ell(\mathcal{T}_S, \bar{Z}_j^i) \mid Z^{1:i-1}]}_{(B)}.$$

Since $\bar{Z}_j^i \sim \bar{Z}_j^{N+1}$ are i.i.d. and independent of $\mathcal{T}_S$ once $Z^{1:N}$ is fixed, we have:

$$\mathbb{E}[\ell(\mathcal{T}_S, \bar{Z}_j^{N+1}) \mid Z^{1:N}] = \mathbb{E}[\ell(\mathcal{T}_S, \bar{Z}_j^i) \mid Z^{1:N}].$$

Hence, (A) = 0.

To control the second term, define a filtration $\mathcal{F}_t := \sigma(Z^1, \ldots, Z^t)$ and define Doob martingale:

$$X_t := \mathbb{E}[\ell(\mathcal{T}_S, \bar{Z}_j^i) \mid Z^1, \ldots, Z^t], \quad t = 0, \ldots, N.$$

Since $\mathcal{T}_S$ is $\beta$-uniformly stable, replacing one sample $Z^t$ changes the expected loss by at most $\beta$. So:

$$|X_t - X_{t-1}| \le \beta.$$

Then, Azuma-Hoeffding implies that with probability at least $1 - \delta$:

$$\left| \mathbb{E}[\ell(\mathcal{T}_S, Z_{i,j}) \mid Z^{1:N}] - \mathbb{E}[\ell(\mathcal{T}_S, Z_{i,j}) \mid Z^{1:i-1}] \right| \le \beta \sqrt{2(N - i + 1) \log(2/\delta)}.$$

Using $\|\mathbf{q}\|_2 \le 1$ and Jensen's inequality:

$$|D_j| \le \sum_{i=1}^N q_i |\Delta_i| \le \|\mathbf{q}\|_2 \cdot \sqrt{\sum_{i=1}^N \beta^2 2(N - i + 1) \log(2/\delta)} \le 2\beta \|\mathbf{q}\| N \sqrt{\log(2/\delta)}.$$

Finally, averaging over $j$:

$$|\mathrm{disc}(\mathbf{q})| \le \frac{1}{N_c} \sum_{j=1}^{N_c} |D_j| \le 2\beta \|\mathbf{q}\|_2 N \sqrt{\log(2/\delta)}.$$

We complete the proof. $\qquad\square$

# F    THE UPPER BOUND ON DISCREPANCY MEASURE UNDER NON-I.I.D SCENARIO

We firstly introduce the definition of Sequential Rademacher utilized in (Rakhlin et al., 2015; Kuznetsov & Mohri, 2015; 2020).

**Definition 7.** *[Sequential Rademacher Complexity] Let* $\boldsymbol{\sigma} = (\sigma_1, \ldots, \sigma_T)$ *be a sequence of Rademacher random variables (each $\sigma_t$ independently taking values $\pm 1$ with equal probability), and let $q = (q_1, \ldots, q_T) \in \mathbb{R}^T$ be a given weight vector. For a function class $\mathcal{G}$ defined on sequential data $z_1, z_2, \ldots, z_T$, the* sequential Rademacher complexity *is*

$$\mathcal{R}_N^{\mathrm{seq}}(\mathcal{G}) := \sup_z \mathbb{E}_{\boldsymbol{\sigma}} \left[ \sup_{g \in \mathcal{G}} \sum_{t=1}^N \sigma_t q_t g(z_t(\boldsymbol{\sigma})) \right],$$

*where the supremum is over all complete (depth-$N$) binary trees or adversarial sequences $z_t(\sigma_1, \ldots, \sigma_{t-1})$.*

In simpler terms, $\mathcal{R}_T^{\mathrm{seq}}(\mathcal{G})$ measures how well $\mathcal{G}$ can fit random signs $\{\sigma_t\}$ in an online or sequential manner.

### F.1 FUNCTION CLASS: TRANSFORMER HYPOTHESIS SPACE

We fix a Transformer architecture (with $N_a$ heads per layer, hidden dimension $D$, and $L$ layers), and let $\theta$ collect all parameters $\{Q_m, K_m, V_m, O_m, W_1, W_2, \dots\}$ across $L$ layers. Denote the overall parameter space by $\Theta$, and suppose we have a norm constraint $\|\theta\| \leq \Lambda$, bounding all weight matrices in operator norm (or some suitable layerwise norm). Let $\mathcal{F}_{\text{Trans}}$ be the function class:

$$\mathcal{F}_{\text{Trans}} := \left\{ f_\theta : \mathbf{P} \mapsto \mathcal{T}(\mathbf{P}) \ \middle| \ \theta \in \Theta, \ \|\theta\| \leq \Lambda \right\}.$$

For *sequential* inputs $\mathbf{P}(\sigma_1, \dots, \sigma_t)$, this means the Transformer is invoked on each partial prompt $\mathbf{P}_{1:t}$.

### F.2 REWRITING THE LOSS AS A COMPOSITE FUNCTION.

Let us set

$$g(\mathbf{P}) \ = \ \mathcal{T}(\mathbf{P})_{n,:} \ \in \ \mathbb{R}^D,$$

and define a function

$$\phi(\mathbf{x}, \mathbf{y}) \ := \ \|\mathbf{x} - \mathbf{y}\|_2^2,$$

where $\mathbf{x}, \mathbf{y} \in \mathbb{R}^D$. Then

$$\ell(\mathcal{T}) \ = \ \phi\big(g(\mathbf{P}), \mathbf{Y}\big) \ = \ \|\mathcal{T}(\mathbf{P})_{n,:} - \mathbf{Y}\|_2^2.$$

Hence the loss class $\mathcal{G} = \{\ell(\mathcal{T})\}$ is precisely $\phi \circ G$ where

$$G \ := \ \big\{ g(\mathbf{P}) = \mathcal{T}(\mathbf{P})_{n,:} \ : \ \mathcal{T} \in \mathcal{H}_{\text{Trans}} \big\}.$$

We check how $\phi(\mathbf{x}, \mathbf{y}) = \|\mathbf{x} - \mathbf{y}\|^2$ depends on $\mathbf{x}$. Let $R = \max\{B_C, \max\{\|\mathbf{H}_{n,:}^L\|\}\}$ be a constant. Suppose $\|\mathbf{x}\| \leq R$ and $\|\mathbf{y}\| \leq R$ for all feasible $(\mathbf{x}, \mathbf{y})$. We can show that $\phi(\mathbf{x}, \mathbf{y})$ is $L_\phi$-Lipschitz in $\mathbf{x}$ with

$$L_\phi \ \leq \ 4 R,$$

because

$$\left| \|\mathbf{x} - \mathbf{y}\|^2 \ - \ \|\mathbf{x}' - \mathbf{y}\|^2 \right| \ \leq \ 4 R \|\mathbf{x} - \mathbf{x}'\|.$$

Thus, if the model outputs $\mathbf{x} = \mathbf{P}$ and targets $\mathbf{Y}$ remain within a ball of radius $RL_\mathcal{T}^*$, then $\phi(\cdot, \mathbf{y})$ is $4RL_\mathcal{T}^*$-Lipschitz in the first coordinate.

### F.3 SEQUENTIAL RADEMACHER COMPLEXITY OF TRANSFORMERS

Inside the expectation $\mathbb{E}_\sigma[\cdot]$, the random variables $\{\sigma_t\}$ are independent Rademacher signs. Let us write:

$$\mathbb{E}_{\boldsymbol{\sigma}}\left[ \sup_{f \in \mathcal{F}} \left| \sum_{t=1}^N \sigma_t \, q_t \, f(z_t) \right| \right] \ \leq \ 4RL_\mathcal{T}^* \, \mathbb{E}_{\boldsymbol{\sigma}}\left[ \left\| \sum_{t=1}^N \sigma_t \, q_t \, z_t \right\| \right].$$

Thus, it remains to bound $\mathbb{E}_{\boldsymbol{\sigma}}\left[ \| \sum_{t=1}^N \sigma_t \, q_t \, z_t \| \right]$. A typical assumption in bounding Rademacher-based complexities is that each $z_t$ has a finite norm $\|z_t\| \leq \sqrt{n} B_P$. Then according to the fact that if $s < t$ then

$$\mathbb{E}_\sigma[\sigma_t \sigma_s q_t q_s z_t z_s] = \mathbb{E}_\sigma[\sigma_t] \mathbb{E}_\sigma[\sigma_s q_t q_s z_t z_s] = 0,$$

we have the following:

$$\mathbb{E}\left[ \| \sum_{t=1}^N \sigma_t \, q_t \, z_t \| \right] \ \leq \ \sqrt{\mathbb{E}\left[ \| \sum_{t=1}^N \sigma_t \, q_t \, z_t \|^2 \right]} \ = \ \sqrt{\sum_{t=1}^N q_t^2 \, \|z_t\|^2} \ \leq \ \sqrt{n} B_P \, \|\mathbf{q}\|_2.$$

Putting all these pieces together:

$$\mathcal{R}_N\big(\{\ell(\mathcal{T})\}\big) \leq \ 4RL_\mathcal{T}^* \sqrt{n} B_P \|\mathbf{q}\|_2.$$

This shows that under norm constraints and bounded inputs/targets, the sequential Rademacher complexity of the squared-$\ell_2$ loss class is finite and depends primarily on the Lipschitz constant of the loss w.r.t. the model's output, as well as on the base complexity of the Transformer itself.

**Remark.** While the above bound may appear loose (e.g. exponential in the number of layers $L$), it nonetheless demonstrates qualitatively that *the capacity of squared-$\ell_2$ losses is controlled* by parameter norms, data magnitude $R$, sequence length $T$, and any submultiplicative structure in the Transformer layers.

For given hypothesis space $\mathcal{H}$ and define by $\hat{\text{disc}}(\mathbf{q}) := \sup_{\mathcal{T} \in \mathcal{H}} \frac{1}{N_c} \sum_{j=1}^{N_c} \left[ E_{N+1,j} - \sum_{i=1}^{N} q_i E_{i,j} \right]$, where $E_{i,j} = \mathbb{E}\left[ \ell(\mathcal{T}(\mathbf{P}^{i,j-1})_{*,:}, C_j^i) | \{(\mathbf{p}^m, \mathbf{c}^m)\}_{m=1}^{i-1} \right]$.

By further combining the fact $\text{disc}(\mathbf{q}) \leq \hat{\text{disc}}(\mathbf{q})$ with the following lemma (Kuznetsov & Mohri, 2020), we obtain the final result.

**Lemma 3.** *For any $\delta > 0$, with probability at least $1 - \delta$, for all $f \in \mathcal{F}$ and all $\alpha > 0$, we have*

$$\sum_{t=1}^{N} \mathbb{E}\left[ q_t f(Z_t) \mid Z_1^{t-1} \right] \leq \sum_{t=1}^{N} q_t f(Z_t) + \|\mathbf{q}\|_2 + 6 M_\ell \sqrt{\pi \log T}\, \mathcal{R}_T(\mathcal{F}) + M_\ell \|\mathbf{q}\|_2 \sqrt{2 \log \frac{1}{\delta}}.$$

Combining this lemma with above results, we complete the proof.

## G    ERROR ACCUMULATION ANALYSIS (PROOF OF THEOREM 5)

In practical scenarios, the model typically uses its own estimated token to predict subsequent tokens. This approach leads to cumulative errors as inaccuracies in steps propagate forward. Denote $\mathcal{L}_i(\mathcal{T}) := \mathcal{L}(\mathcal{T}(\mathbf{P}^{i,j-1}), C_j^i)$ the population risk at $i$-th step prediction such that $\mathcal{L}(\mathcal{T}) = \frac{1}{N_c} \sum_{i=1}^{N_c} \mathcal{L}_i(\mathcal{T})$. We have the following relation between $\mathcal{L}^{EA}(\mathcal{T})$ and $\mathcal{L}(\mathcal{T})$.

**Theorem 5.** *(Error Accumulation Analysis) Let $\mathcal{L}(\mathcal{T})$ and $\mathcal{L}^{EA}(\mathcal{T})$ be defined in Eqs (1) - (2). Assume the conditions in Theorem 4 hold. For any $0 < \delta < 1$, we have*

$$\mathcal{L}^{EA}(\mathcal{T}_S) \leq \frac{1}{N_c} \mathcal{L}_{N_c} + \frac{L_{\mathcal{T}}^*}{N_c} \sum_{j=1}^{N_c-1} \left[ \frac{1}{L_{\mathcal{T}}^*} + \delta_{j=1} \prod_{i=2}^{N_c-1} (1 + \frac{L_{\mathcal{T}}^*}{i}) + \sum_{i=j+1}^{N_c-1} \prod_{k=i+1}^{N_c-1} (1 + \frac{L_{\mathcal{T}}^*}{k}) \right] \mathcal{L}_j,$$

*where $\delta_{(.)}$ is the Kronecker delta.*

It is evident that $L_{\mathcal{T}}^*$ (defined in Eq. (12)) increases with the number of layers $L$, and polynomially with the prompt length, as well as linearly with the model size parameters $N_a$ and $D$. Based on this observation, we focus on analyzing the impact of inference length on the generalization error bound.

**Corollary 5.** *[Generalisation under i.i.d Scenario] Let $|B| = O(N^{\zeta_1})$, $\zeta_1 \leq \frac{1}{2}$, $Q = O(\ln N)$ and $N_c = O((\ln N)^{\zeta_2})$. With at least confidence $1 - \delta$, there holds*

$$\mathcal{L}^{EA}(\mathcal{T}_S) \lesssim \sum_{i=1}^{N} \sum_{j=1}^{N_c} q_i \eta_j \ell(\mathcal{T}_S(\mathbf{p}^i)_{*,:}, c_j^i) + \psi \sqrt{2 \log \frac{4}{\delta}},$$

*where $\psi = (\log n)^{\zeta_2 (\log n)^{\zeta_2 2L}} N^{-1/2}$, and the weights $\eta_j, j = 1, \ldots, N_c$ equal to*

$$\frac{L_{\mathcal{T}}^*}{N_c} \left[ \frac{1}{L_{\mathcal{T}}^*} + \delta_{j=1} \prod_{i=2}^{N_c-1} (1 + \frac{L_{\mathcal{T}}^*}{i}) + \sum_{i=j+1}^{N_c-1} \prod_{k=i+1}^{N_c-1} (1 + \frac{L_{\mathcal{T}}^*}{k}) \right].$$

This result suggests, in scenarios where error accumulation occurs, the length of the inference process should be constrained to a logarithmic scale relative to the sample size to ensure effective generalisation. Notably, this finding aligns with (Merrill & Sabharwal, 2023), which states that transformers with a logarithmic number of intermediate tokens may exhibit enhanced computational power. Similar results can be easily extended to the non-i.i.d. setting.

*Proof.* According to the definitions of $\mathcal{L}(\theta)$ and $\mathcal{L}^{EA}(\theta)$, we have the follows

$$
\begin{aligned}
\mathcal{L}^{EA}(\theta) - \mathcal{L}(\theta) &= \frac{1}{N_c}\sum_{i=1}^{N_c}\mathcal{L}_i^{EA}(\theta) - \frac{1}{N_c}\sum_{i=1}^{N_c}\mathcal{L}_i(\theta) \\
&= \frac{1}{N_c}\sum_{j=1}^{N_c}\mathbb{E}[\ell(\mathcal{T}(\hat{\mathbf{P}}^{N+1,j-1})_{*,:},C_j^{N+1})] - \mathbb{E}[\ell(\mathcal{T}(\mathbf{P}^{N+1,j-1})_{*,:},C_j^{N+1})] \\
&= \frac{1}{N_c}\sum_{j=1}^{N_c}\mathbb{E}[\|\mathcal{T}(\mathbf{P}^{N+1,j-1})_{*,:} - \mathcal{T}(\hat{\mathbf{P}}^{N+1,j-1})_{*,:}\|_2^2].
\end{aligned}
$$

According to the Lipschitz property of Transformer, we have

$$
\begin{aligned}
\frac{1}{N_c}\sum_{j=1}^{N_c}\mathbb{E}[\|\mathcal{T}(\mathbf{P}^{N+1,j-1}) - \mathcal{T}(\hat{\mathbf{P}}^{N+1,j-1})\|_2^2] &\leq \frac{L_{\mathcal{T}}^*}{N_c}\sum_{j=1}^{N_c}\mathbb{E}[\|\mathbf{P}^{N+1,j-1} - \hat{\mathbf{P}}^{N+1,j-1}\|_2^2] \\
&= \frac{L_{\mathcal{T}}^*}{N_c}\sum_{i=1}^{N_c}\sum_{j=1}^{i}\mathbb{E}[\|\mathcal{T}(\hat{\mathbf{P}}^{N+1,j-1}) - C_j^{N+1}\|_2^2] \\
&= \frac{L_{\mathcal{T}}^*}{N_c}\sum_{i=1}^{N_c-1}\sum_{j=1}^{i}\mathcal{L}_j^{EA}(\theta),
\end{aligned}
$$

For notational simplicity, we denote $A_i^{EA} = \sum_{j=1}^{i}\mathcal{L}_j^{EA}(\theta)$ and $A_i = \sum_{j=1}^{i}\mathcal{L}_j(\theta)$. Then we have

$$
A_m^{EA} \leq A_m + \frac{L_{\mathcal{T}}^*}{m}\sum_{i=1}^{m-1}A_i^{EA}.
$$

We denote by $S_n = \sum_{i=1}^{n}A_i^{EA}$. Since

$$
\begin{aligned}
S_n = S_{n-1} + A_n^{EA} &\leq (1 + \frac{L_{\mathcal{T}}^*}{n})S_{n-1} + A_n \\
&\leq \prod_{i=2}^{n}(1+\frac{L_{\mathcal{T}}^*}{i})A_1 + \sum_{i=2}^{n}\prod_{k=i+1}^{n}(1+\frac{L_{\mathcal{T}}^*}{k})A_i \\
&\leq \sum_{j=1}^{n}\Big[\delta_{j=1}\prod_{i=2}^{n}(1+\frac{L_{\mathcal{T}}^*}{i}) + \sum_{i=j+1}^{n}\prod_{k=i+1}^{n}(1+\frac{L_{\mathcal{T}}^*}{k})\Big]\mathcal{L}_j.
\end{aligned}
$$

Thus, we have

$$
\begin{aligned}
\mathcal{L}^{EA}(\theta) &\leq \mathcal{L}(\theta) + \frac{L_{\mathcal{T}}^*}{N_c}\sum_{i=1}^{N_c-1}A_i^{EA}(\theta) \\
&\leq \frac{1}{N_c}\sum_{i=1}^{N_c}\mathcal{L}_i + \frac{L_{\mathcal{T}}^*}{N_c}\sum_{j=1}^{N_c-1}\Big[\delta_{j=1}\prod_{i=2}^{N_c-1}(1+\frac{L_{\mathcal{T}}^*}{i}) + \sum_{i=j+1}^{N_c-1}\prod_{k=i+1}^{N_c-1}(1+\frac{L_{\mathcal{T}}^*}{k})\Big]\mathcal{L}_j \\
&\leq \frac{1}{N_c}\mathcal{L}_{N_c} + \frac{L_{\mathcal{T}}^*}{N_c}\sum_{j=1}^{N_c-1}\Big[\frac{1}{L_{\mathcal{T}}^*} + \delta_{j=1}\prod_{i=2}^{N_c-1}(1+\frac{L_{\mathcal{T}}^*}{i}) + \sum_{i=j+1}^{N_c-1}\prod_{k=i+1}^{N_c-1}(1+\frac{L_{\mathcal{T}}^*}{k})\Big]\mathcal{L}_j.
\end{aligned}
$$

This completes the proof.

We then simplify these weights. Using the logarithmic approximation, there holds

$$
\prod_{i=2}^{N_c-1}\left(1+\frac{L_{\mathcal{T}}^*}{i}\right) \approx \exp\left(L_{\mathcal{T}}^*\sum_{i=2}^{N_c-1}\frac{1}{i}\right) \leq N_c^{L_{\mathcal{T}}^*}.
$$

Similarly,

$$\prod_{k=i+1}^{N_c-1} \left(1 + \frac{L_{\mathcal{T}}^*}{k}\right) \le \left(\frac{N_c}{i+1}\right)^{L_{\mathcal{T}}^*}.$$

Approximating the summation, we have

$$\sum_{i=j+1}^{N_c-1} \left(\frac{N_c}{i+1}\right)^{L_{\mathcal{T}}^*} \approx \int_j^{N_c} \left(\frac{N_c}{x}\right)^{L_{\mathcal{T}}^*} dx.$$

Evaluating the integral yields

$$\frac{N_c^{L_{\mathcal{T}}}}{L_{\mathcal{T}}} \left[(N_c^{-L_{\mathcal{T}}^*+1} - j^{-L_{\mathcal{T}}^*+1})\right] = \frac{N_c^{L_{\mathcal{T}}^*} - j^{L_{\mathcal{T}}^*}}{L_{\mathcal{T}}^*}.$$

Thus, the dominant term in the simplified bound is:

$$O\left(L_{\mathcal{T}}^* N_c^{L_{\mathcal{T}}^*-1} - j^{L_{\mathcal{T}}^*-1}\right).$$

By combining above results with the generalisation bound established above, we obtain Corollary 5. $\qquad\square$

## H   GRADIENT, HESSIAN MATRIX AND LIPSCHITZ (SMOOTH) CONSTANT

This section gives the gradient and hessian of Transformer models. Note that, $n, N_p \le n \le N_p + N_c$ in this section refers to the length of input prompt.

### H.1   GRADIENT W.R.T. $\mathbf{H}_{n,:}^l$ AND ITS NORM UPPER BOUND

It is easy to obtain the gradient of loss w.r.t. final output $\mathbf{H}_{n,:}^L$, i.e.,

$$\frac{\partial \mathcal{L}}{\partial \mathbf{H}_{*,:}^L} = 2(\mathbf{H}_{*,:}^L - \mathbf{y}).$$

To establish the upped bound on $\|\frac{\partial \mathcal{L}}{\partial \mathbf{H}_{*,:}^L}\|_2$, we need to bound the upper bound on the output $\|\mathbf{H}_{*,:}^L\|_2$. The definition of Transformer models yields

$$\mathbf{H}_{*,:}^L = \mathcal{M}^L(\mathcal{A}^L(\mathbf{H}^{L-1}))_{*,:} = \mathrm{ReLU}(\mathcal{A}^L \mathbf{H}_{*,:}^{L-1} W_1^L) W_2^L$$

and

$$\|\mathcal{M}^L(\mathcal{A}^L(\mathbf{H}^{L-1}))_{i,:}\|_2 \le B_{W_1} B_{W_2} \sup \|\mathcal{A}^L \mathbf{H}_{i,:}^{L-1}\|_2,$$

We have

$$
\begin{aligned}
\|\mathcal{A}^L(\mathbf{H}^{L-1}))_{n,:}\|_2 &= \|\sum_{m=1}^{N_a} \mathrm{softmax}\big((\mathbf{H}^{L-1})_{i,:} Q_m^L K_m^L (\mathbf{H}^{L-1})^T\big) \mathbf{H}^{L-1} V_m^L O_m^L\|_2 \\
&\le B_V B_O N_a \sum_{j=1}^n s_j \|\mathbf{H}_{j,:}^{L-1}\|_2 \\
&\le B_V B_O N_a \sup_{j=1,\ldots,n} \|\mathbf{H}_{j,:}^{L-1}\|_2.
\end{aligned}
$$

Combined with above result, we obtain

$$\sup \|\mathbf{H}_{i,:}^L\|_2 \le B_{W_1} B_{W_2} B_V B_O N_a \sup_{i=1,\ldots,n} \|\mathbf{H}_{i,:}^{L-1}\|_2 \le (B_{W_1} B_{W_2} B_V B_O N_a)^L \sup_{i=1,\ldots,n} \|\mathbf{H}_{i,:}^0\|_2.$$

Under Assumption 1, we have

$$\sup \|2(\mathbf{H}_{n,:}^L - \mathbf{y})\|_2 \le \sqrt{2}(B_{W_1} B_{W_2} B_V B_O N_a)^L B_P + \sqrt{2} B_C =: C_L$$

Similarly, the maximum of loss function can be bounded by

$$M_\ell = 2(B_{W_1} B_{W_2} B_V B_O N_a)^{2L} B_P + 2B_C^2.$$

## H.2   GRADIENT W.R.T. $W_2^l$ AND ITS NORM UPPER BOUND

For any $l = 1, ..., L$, the gradient w.r.t. $W_2^l$, we get

$$\frac{\partial \mathcal{L}}{\partial W_2^l} = \frac{\partial \mathcal{L}}{\partial \mathbf{H}_{n,:}^L} \Big[ \frac{\partial \mathbf{H}_{n,:}^L}{\partial \mathbf{H}^{L-1}} \cdots \frac{\partial \mathbf{H}^{l+1}}{\partial \mathbf{H}^l} \Big] \frac{\partial \mathbf{H}^l}{\partial W_2^l}.$$

According to the definition of $\mathcal{M}$ and $\mathcal{A}$, we have

$$\frac{\partial \mathbf{H}^l}{\partial W_2^l} = \text{ReLU}(\mathcal{A}^l(\mathbf{H}^{l-1})W_1^l).$$

Recalling the Lipschitz property of Transformer, we have

$$\|\frac{\partial \mathbf{H}_{n,:}^L}{\partial \mathbf{H}^{L-1}} \cdots \frac{\partial \mathbf{H}^{l+1}}{\partial \mathbf{H}^l}\|_F \leq n^{-1} L_{\mathcal{T}}^{\frac{L-l}{L}} := C_{l:L}$$

We then can bound the gradient w.r.t $W_2^l$ by

$$\|\frac{\partial \mathcal{L}}{\partial W_2^l}\|_F \leq C_L C_{l:L} B_{W_1} \|\mathcal{A}^l(\mathbf{H}^{l-1})\|_F,$$

where

$$\|\mathcal{A}^l(\mathbf{H}^{l-1})\|_F \quad \leq \quad \sqrt{n} \sup_{i=1,...,n} \|\mathcal{A}^l(\mathbf{H}^{l-1})_{i,:}\|_F$$

$$\leq \quad \sqrt{n} B_V B_O N_a \sup_{i=1,...,n} \|\mathbf{H}_{i,:}^{l-1}\|_F \leq \sqrt{n} B_V^l B_O^l N_a^l (B_{W_1} B_{W_2})^{l-1} B_P := C_{W_2}.$$

## H.3   GRADIENT W.R.T. $W_1^l$ AND ITS NORM UPPER BOUND

We next to give the upper bound on the norm of the gradient w.r.t. $W_1^l$. Similarly, for any $l = 1, ..., L$, we have

$$\frac{\partial \mathcal{L}}{\partial W_1^l} = \frac{\partial \mathcal{L}}{\partial \mathbf{H}_{n,:}^L} \Big[ \frac{\partial \mathbf{H}_{n,:}^L}{\partial \mathbf{H}^{L-1}} \cdots \frac{\partial \mathbf{H}^{l+1}}{\partial \mathbf{H}^l} \Big] \frac{\partial \mathbf{H}^l}{\partial W_1^l}.$$

and

$$\frac{\partial \mathbf{H}^l}{\partial W_1^l} = \Big[\mathbf{R}^l \odot \frac{\partial \mathcal{A}^l(\mathbf{H}^{l-1})W_1^l}{\partial W_1^l}\Big] W_2^l = [\mathbf{R}^l \odot \mathcal{A}^l(\mathbf{H}^{l-1})] W_2^l,$$

where $R_{ij}^l = 1$ if $(\mathcal{A}^l(\mathbf{H}^{l-1})W_1^l)_{ij} > 0$, otherwise $R_{ij} = 0$. The upper bound is

$$\|\frac{\partial \mathcal{L}}{\partial W_1^l}\|_F \leq C_L C_{l:L} B_{W_2} \|\mathbf{R}^l \odot \mathcal{A}^l(\mathbf{H}^{l-1})\|_F \leq C_L C_{l:L} B_{W_2} \|\mathcal{A}^l(\mathbf{H}^{l-1})\|_F.$$

## H.4   GRADIENT W.R.T. $Q_m^l$ AND ITS NORM UPPER BOUND

For any $l = 1, ..., L$, the gradient w.r.t. $Q_m^l$ is

$$\frac{\partial \mathcal{L}}{\partial Q_m^l} = \frac{\partial \mathcal{L}}{\partial \mathbf{H}_{n,:}^L} \Big[ \frac{\partial \mathbf{H}_{n,:}^L}{\partial \mathbf{H}^{L-1}} \cdots \frac{\partial \mathbf{H}^{l+1}}{\partial \mathbf{H}^l} \Big] \frac{\partial \mathbf{H}^l}{\partial Q_m^l}$$

and

$$\frac{\partial \mathbf{H}^l}{\partial Q_m^l} = \frac{\partial \mathbf{H}^l}{\partial \mathcal{A}^l(\mathbf{H}^{l-1})} \frac{\partial \mathcal{A}^l(\mathbf{H}^{l-1})}{\partial Q_m^l} = \begin{pmatrix} \text{diag}(\mathbf{R}_1^l) W_1^l W_2^l \\ \vdots \\ \text{diag}(\mathbf{R}_n^l) W_1^l W_2^l \end{pmatrix} \frac{\partial \mathcal{A}^l(\mathbf{H}^{l-1})}{\partial Q_m^l},$$

where

$$\frac{\partial \mathcal{A}^l(\mathbf{H}^{l-1})}{\partial Q_m^l} = \text{softmax}' \cdot (\mathbf{H}^{l-1})^\top \mathbf{H}^{l-1} K_m^l \mathbf{H}^{l-1} V_m^l O_m^l$$

and

$$\text{softmax}' = \text{softmax}(Z)(\mathbf{I} - \text{softmax}(Z)^T).$$

Note that $Z = \mathbf{H}^{l-1} Q_m^l K_m^l (\mathbf{H}^{l-1})^T$ and

$$\|\text{softmax}'\|_F \le \frac{\sqrt{n}}{2}.$$

Then the corresponding upper bound is

$$
\begin{aligned}
\|\frac{\partial \mathcal{L}}{\partial Q_m^l}\|_F &\le C_L C_{l:L} \sqrt{n} B_{W_1} B_{W_2} \|\frac{\partial \mathcal{A}^l(\mathbf{H}^{l-1})}{\partial Q_m^l}\|_F \\
&\le C_L C_{l:L} n B_{W_1} B_{W_2} B_K B_V B_O \|\mathbf{H}^{l-1}\|_2^{3/2} \\
&\le C_L C_{l:L} n^2 B_{W_1}^{\frac{3l-1}{2}} B_{W_2}^{\frac{3l-1}{2}} B_K B_V^{\frac{3l-1}{2}} B_O^{\frac{3l-1}{2}} N_a^{\frac{3(l-1)}{2}} B_P^{\frac{3}{2}}.
\end{aligned}
$$

## H.5   GRADIENT W.R.T. $K_m^l$ AND ITS NORM UPPER BOUND

Similarly, the corresponding upper bound of the norm of the gradient w.r.t. $K_m^l$ is

$$
\begin{aligned}
\left\|\frac{\partial \mathbf{H}^l}{\partial K_m^l}\right\|_F &\le C_L C_{l:L} \sqrt{n} B_{W_1} B_{W_2} \left\|\frac{\partial \mathcal{A}^l(\mathbf{H}^{l-1})}{\partial K_m^l}\right\|_F \\
&\le C_L C_{l:L} n^2 B_{W_1}^{\frac{3l-1}{2}} B_{W_2}^{\frac{3l-1}{2}} B_Q B_V^{\frac{3l-1}{2}} B_O^{\frac{3l-1}{2}} N_a^{\frac{3(l-1)}{2}} B_P^{\frac{3}{2}}.
\end{aligned}
$$

## H.6   GRADIENT W.R.T. $V_m^l$ AND ITS NORM UPPER BOUND

For any $l = 1, ..., L$, the gradient w.r.t. $V_m^l$ is

$$\frac{\partial \mathcal{L}}{\partial V_m^l} = \frac{\partial \mathcal{L}}{\partial \mathbf{H}_{n,:}^L}\left[\frac{\partial \mathbf{H}_{n,:}^L}{\partial \mathbf{H}^{L-1}} \cdots \frac{\partial \mathbf{H}^{l+1}}{\partial \mathbf{H}^l}\right]\frac{\partial \mathbf{H}^l}{\partial V_m^l}.$$

and

$$
\frac{\partial \mathbf{H}^l}{\partial V_m^l} = \frac{\partial \mathbf{H}^l}{\partial \mathcal{A}^l(\mathbf{H}^{l-1})}\frac{\partial \mathcal{A}^l(\mathbf{H}^{l-1})}{\partial V_m^l} = \begin{pmatrix} \text{diag}(\mathbf{R}_1^l)W_1^l W_2^l \\ \vdots \\ \text{diag}(\mathbf{R}_n^l)W_1^l W_2^l \end{pmatrix}\frac{\partial \mathcal{A}^l(\mathbf{H}^{l-1})}{\partial V_m^l},
$$

where

$$\frac{\partial \mathcal{A}^l(\mathbf{H}^{l-1})}{\partial V_m^l} = \text{softmax}\left(\mathbf{H}^{l-1} Q_m^l (\mathbf{H}^{l-1} K_m^l)^\top\right)^\top \mathbf{H}^{l-1} O_m^l.$$

Then we can bound $\|\frac{\partial \mathbf{H}^l}{\partial V_m^l}\|_F$ by

$$
\begin{aligned}
\|\frac{\partial \mathcal{L}}{\partial V_m^l}\|_F &\le C_L C_{l:L} \sqrt{n} B_{W_1} B_{W_2} \|\frac{\partial \mathcal{A}^l(\mathbf{H}^{l-1})}{\partial V_m^l}\|_F \\
&\le C_L C_{l:L} n B_{W_1} B_{W_2} B_O \|\mathbf{H}^{l-1}\|_F \\
&\le C_L C_{l:L} n^{\frac{3}{2}} B_{W_1} B_{W_2} B_O \|\mathbf{H}_{n,:}^{l-1}\|_F \\
&\le C_L C_{l:L} n^{\frac{3}{2}} B_{W_1}^l B_{W_2}^l B_O^l B_V^{l-1} N_a^{l-1} B_P.
\end{aligned}
$$

## H.7   GRADIENT W.R.T. $O_m^l$ AND ITS NORM UPPER BOUND

Similarly, there also holds

$$\frac{\partial \mathcal{L}}{\partial O_m^l} = \frac{\partial \mathcal{L}}{\partial \mathbf{H}_{n,:}^L}\left[\frac{\partial \mathbf{H}_{n,:}^L}{\partial \mathbf{H}^{L-1}} \cdots \frac{\partial \mathbf{H}^{l+1}}{\partial \mathbf{H}^l}\right]\frac{\partial \mathbf{H}^l}{\partial O_m^l}$$

and

$$
\frac{\partial \mathbf{H}^l}{\partial O_m^l} = \frac{\partial \mathbf{H}^l}{\partial \mathcal{A}^l(\mathbf{H}^{l-1})}\frac{\partial \mathcal{A}^l(\mathbf{H}^{l-1})}{\partial O_m^l} = \begin{pmatrix} \text{diag}(\mathbf{R}_1^l)W_1^l W_2^l \\ \vdots \\ \text{diag}(\mathbf{R}_n^l)W_1^l W_2^l \end{pmatrix}\frac{\partial \mathcal{A}^l(\mathbf{H}^{l-1})}{\partial O_m^l},
$$

where

$$\frac{\partial \mathcal{A}^l(\mathbf{H}^{l-1})}{\partial O_m^l} = \text{softmax}\left(\mathbf{H}^{l-1}Q_m^l(\mathbf{H}^{l-1}K_m^l)^\top\right)^\top \mathbf{H}^{l-1}V_m^l.$$

Then we can bound $\|\frac{\partial \mathbf{H}^l}{\partial O_m^l}\|_F$ by

$$\begin{aligned}
\|\frac{\partial \mathcal{L}}{\partial O_m^l}\|_F &\leq C_L C_{l:L}\sqrt{n}B_{W_1}B_{W_2}\|\frac{\partial \mathcal{A}^l(\mathbf{H}^{l-1})}{\partial O_m^l}\|_F \\
&\leq C_L C_{l:L}nB_{W_1}B_{W_2}B_V\|\mathbf{H}^{l-1}\|_F \\
&\leq C_L C_{l:L}n^{\frac{3}{2}}B_{W_1}B_{W_2}B_V\|\mathbf{H}_{n,:}^{l-1}\|_F \\
&\leq C_L C_{l:L}n^{\frac{3}{2}}B_{W_1}^l B_{W_2}^l B_V^l B_O^{l-1} N_a^{l-1} B_P.
\end{aligned}$$

## H.8 HESSIAN MATRIX

We firstly calculate the Hessian of Transformer $T^l$ w.r.t $\mathbf{H}^{l-1}$ and its upper bound.

For $l$-th layer, the Hessian matrix is

$$H_{T^l} = \nabla_{\mathbf{H}^{l-1}}^2 \text{ReLU}(\mathcal{A}^l(\mathbf{H}^{l-1})W_1^l)W_2^l = \text{diag}(\text{ReLU}'(\mathcal{A}^l(\mathbf{H}^{l-1})W_1^l))\nabla_{\mathbf{H}^{l-1}}^2 \mathcal{A}^l(\mathbf{H}^{l-1})W_1^l W_2^l.$$

where

$$\nabla_{\mathbf{H}^{l-1}}^2 \mathcal{A}^l = \sum_{m=1}^{N_a} \left(\nabla_{Z_m}^2 \text{softmax}(Z_m) \cdot (\nabla_{\mathbf{H}^{l-1}}Z_m)^2 \cdot V_m^l O_m^l + \nabla_{Z_m}\text{softmax}(Z_m) \cdot \nabla_{\mathbf{H}^{l-1}}^2 Z_m \cdot V_m^l O_m^l\right)$$

and

$$Z_m = \mathbf{H}^{l-1}Q_m^l K_m^l(\mathbf{H}^{l-1})^T.$$

The norm of each component in Lipschitz smooth constant for $T^l$ w.r.t $\mathbf{H}^{l-1}$ is

$$\|\text{diag}(\text{ReLU}'(\mathcal{A}^l(\mathbf{H}^{l-1})W_1^l))\| \leq \sqrt{nD},$$

$$\nabla_{\mathbf{H}^{l-1}}^2 \mathcal{A}^l(\mathbf{H}^{l-1}) \leq C_{softmax}N_a B_V B_O B_Q B_K \|\mathbf{H}^{l-1}\|_F(B_Q B_K\|\mathbf{H}^{l-1}\|_F + 2),$$

where a conservative bound $C_{softmax} = D^2/8$. Then we have

$$\|H_{T^l}\|_F \leq \sqrt{n}D^{\frac{3}{2}}N_a B_V B_O B_Q B_K B_{W_1}B_{W_2}\|\mathbf{H}^{l-1}\|_F(\|\mathbf{H}^{l-1}\|_F + 2).$$

For $W_1^l$ and $W_2^l$, their Lipschitz smooth constants are 0. For $Q_m^l$, we have the Hessian matrix

$$H_{Q_m^l} = \nabla_{Q_m^l}^2 \text{ReLU}(\mathcal{A}^l(\mathbf{H}^{l-1})W_1^l)W_2^l = \text{diag}(\text{ReLU}'(\mathcal{A}^l(\mathbf{H}^{l-1})W_1^l))\nabla_{Q_m^l}^2 \mathcal{A}^l(\mathbf{H}^{l-1})W_1^l W_2^l.$$

where

$$\nabla_{Q_m^l}^2 \mathcal{A}^l = \sum_{m=1}^{N_a} \left(\nabla_{Z_m}^2 \text{softmax}(Z_m) \cdot (\nabla_{Q_m^l}Z_m)^2 \cdot V_m^l O_m^l + \nabla_{Z_m}\text{softmax}(Z_m) \cdot \nabla_{Q_m^l}^2 Z_m \cdot V_m^l O_m^l\right)$$

and

$$\|\nabla_{Q_m^l}^2 \mathcal{A}^l\|_F \leq N_a D^2 B_V B_O B_K^2 \|\mathbf{H}^{l-1}\|^2.$$

For $K_m^l$, we similarly have

$$\|\nabla_{K_m^l}^2 \mathcal{A}^l\|_F \leq N_a D^2 B_V B_O B_Q^2 \|\mathbf{H}^{l-1}\|^2.$$

For $V_m^l$ and $O_m^l$, the Lipschitz smooth constants are 0.

## H.9 Lipschitz Constant $L_\ell$, Lipschitz Smooth Constant $\gamma$ and Maximum of Loss Function $M_\ell$

According to the results in Appendix H, the upper bound on $L_\ell$, $M_\ell$ and $\gamma$ are

$$M_\ell = 2(B_{W_1} B_{W_2} B_V B_O N_a)^{2L} B_P + 2B_C^2, \tag{7}$$

$$L_\ell = \sum_{n=N_e N_c+1}^{(N_e+1)N_c} v_n \sum_{l=1}^{L} (C_{W_1}^l + C_{W_2}^l + N_a(C_Q^l + C_K^l + C_V^l + C_O^l)) \tag{8}$$

where

$$C_{W_1}^l = C_L C_{l:L} B_{W_2} \sqrt{n} B_V^l B_O^l N_a^l (B_{W_1} B_{W_2})^{l-1} B_P$$

$$C_{W_2}^l = C_L C_{l:L} B_{W_1} \sqrt{n} B_V^l B_O^l N_a^l (B_{W_1} B_{W_2})^{l-1} B_P$$

$$C_Q^l = C_L C_{l:L} n^2 B_{W_1}^{\frac{3l-1}{2}} B_{W_2}^{\frac{3l-1}{2}} B_K B_V^{\frac{3l-1}{2}} B_O^{\frac{3l-1}{2}} N_a^{\frac{3(l-1)}{2}} B_P^{\frac{3}{2}}.$$

$$C_K^l = C_L C_{l:L} n^2 B_{W_1}^{\frac{3l-1}{2}} B_{W_2}^{\frac{3l-1}{2}} B_Q B_V^{\frac{3l-1}{2}} B_O^{\frac{3l-1}{2}} N_a^{\frac{3(l-1)}{2}} B_P^{\frac{3}{2}}$$

$$C_V^l = C_L C_{l:L} n^{\frac{3}{2}} B_{W_1}^l B_{W_2}^l B_O^l B_V^{l-1} N_a^{l-1} B_P.$$

$$C_O^l = C_L C_{l:L} n^{\frac{3}{2}} B_{W_1}^l B_{W_2}^l B_V^l B_O^{l-1} N_a^{l-1} B_P.$$

The Lipschitz smooth constant is

$$\gamma = \sum_{n=N_e N_c+1}^{(N_e+1)N_c} v_n \sum_{l=1}^{L} N_a(\gamma_Q^l + \gamma_K^l) \tag{9}$$

where

$$\gamma_Q^l = \left( \sqrt{n} D^{\frac{3}{2}} N_a B_V B_O B_Q B_K B_{W_1} B_{W_2} \|\mathbf{H}^{l-1}\|_F (\|\mathbf{H}^{l-1}\|_F + 2) \right)^{L-l} N_a D^2 B_V B_O B_K^2 \|\mathbf{H}^{l-1}\|^2$$

and

$$\gamma_K^l = \left( \sqrt{n} D^{\frac{3}{2}} N_a B_V B_O B_Q B_K B_{W_1} B_{W_2} \|\mathbf{H}^{l-1}\|_F (\|\mathbf{H}^{l-1}\|_F + 2) \right)^{L-l} N_a D^2 B_V B_O B_Q^2 \|\mathbf{H}^{l-1}\|^2,$$

where

$$\|\mathbf{H}^{l-1}\|_F \le n^{\frac{1}{2}} (B_{W_1} B_{W_2} B_V B_O N_a)^{l-1} B_P.$$

Putting the above results into the upper bound of stability will obtain the desirable results.

## H.10 Lipschitz Constant w.r.t Input $L_{\mathcal{T}}$

In a Transformer, the ReLU activation function is piecewise linear and thus non-differentiable at certain points. In particular, the concept of a Jacobian, defined in terms of the network's outputs relative to its inputs, indicates how those outputs vary with small changes in the inputs. The Jacobian at a point $x$ is computed via the chain rule during backpropagation. However, it is only well-defined if all ReLU nodes are differentiable at that point, meaning their inputs must be strictly positive or strictly negative. Consequently, if an input equals zero, one must assume the existence of a sub-gradient within $[0, 1]$.

According to the chain rule, the Jacobian at a point $\mathbf{p}$ (namely $\mathbf{H}^0$), if defined, can be compactly represented as:

$$J_{\mathbf{p}}[\mathcal{T}] = J_{\mathbf{p}}[T^L \circ T^{L-1} \circ \cdots \circ T^1] = J_{\mathbf{H}^{L-1}}[T^L] \cdots J_{\mathbf{H}^0}[T^1].$$

To obtain $J_{\mathbf{H}^{l-1}}[T^l], l = 1, ..., L$, we need to calculate $J_{\mathcal{A}^l(\mathbf{H}^{l-1})}[\mathcal{M}^l]$ and $J_{\mathbf{H}^{l-1}}[\mathcal{A}^l]$, respectively. Since both $\mathcal{M}^l$ and $\mathcal{A}^l$ map from $\mathbb{R}^{N_p \times D}$ to $\mathbb{R}^{N_p \times D}$, their Jacobian matrices have the same form

$$
\begin{bmatrix}
J_{11}^l & J_{12}^l & \cdots & J_{1N_p}^l \\
J_{21}^l & J_{22}^l & \cdots & J_{2N_p}^l \\
\vdots & \vdots & \ddots & \vdots \\
J_{N_p 1}^l & J_{N_p 2}^l & \cdots & J_{N_p N_p}^l
\end{bmatrix}
\in \mathbb{R}^{N_p D \times N_p D}
\tag{10}
$$

We firstly give the Jacobian matrix for $J_{\mathbf{A}^l}[\mathcal{M}^l]$. We denote by $\mathbf{A}^l = \mathcal{A}^l(\mathbf{H}^{l-1}) \in \mathbb{R}^{N_p \times D}$. Recall the definition of mapping

$$
\mathcal{M}^l(\mathbf{A}^l) = \mathrm{ReLU}(\mathbf{A}^l W_1^l) W_2^l =
\begin{bmatrix}
M_1^T(\mathbf{A}^l) \\
M_2^T(\mathbf{A}^l) \\
\vdots \\
M_N^T(\mathbf{A}^l)
\end{bmatrix}
\in \mathbb{R}^{N_p \times D},
$$

where $M_i(\mathbf{A}^l) = \mathrm{ReLU}(\mathbf{A}_i^l W_1^l) W_2^l$. By taking partial derivatives, for any $i, j \in [N_p]$, we have

$$
J_{ij}^l = \frac{\partial M_i(\mathbf{A}^l)}{\partial \mathbf{A}_j^l} = \frac{\partial \mathrm{ReLU}(\mathbf{A}_i^l W_1) W_2}{\partial \mathbf{A}_j^l} = \delta_{ij} W_2^T W_1^T \mathcal{G}_i^l,
$$

where $\mathcal{G}^l$ encodes the activation pattern of a layer $l$ caused by the input $x$, and $\delta_{ij}$ is the Kronecker delta. The matrix $\mathcal{G}_i^l$ is a diagonal matrix, having 1s as elements if the corresponding neuron is active, otherwise 0s for inactive neurons. The Jacobian is the same for all the points strictly inside a linear region with the same activation pattern. Since ReLU networks are piece-wise linear in nature, the Lipschitz constant is exactly equal to the $p$-norm of the Jacobian at one such linear region in the input domain. Thus, the jacobian matrix for $\mathcal{M}$ is a diagonal block matrix, having $W_2^T W_1^T \mathcal{G}_i^l$ as elements if $i = j$, otherwise $0^{D \times D}$ for $i \neq j$. We then have

$$
\sup_{\mathbf{A}^l} \| J_{\mathbf{A}^l}[\mathcal{M}^l] \|_2 \leq N_p^{1/2} D B_{W_1} B_{W_2}.
$$

In fact, there have been some studies analyzed $J_{\mathbf{H}^{l-1}}[\mathcal{A}^l], l = 1, ..., L$ and the Lipschitz constant of attention . Since transformer $\mathcal{T}$ is a map from $\mathbb{R}^{N_p \times D}$ to $\mathbb{R}^{N_p \times D}$, the element of Jacobian is $J_{ij}^l = \frac{\partial (\mathcal{A}^l(\mathbf{H}^{l-1}))_i}{\partial \mathbf{H}_j^{l-1}}$. The Jacobian of the softmax is also well-known. Suppose that $\mathbf{v} = \mathrm{softmax}(\mathbf{u}) \in \mathbb{R}^{N_p \times 1}$. Then we have

$$
\frac{\partial \mathbf{v}}{\partial \mathbf{u}} = \mathrm{diag}(\mathbf{v}) - \mathbf{v}\mathbf{v}^T
$$

Recall the definition of mapping

$$
\mathcal{A}^l(\mathbf{H}^{l-1}) \quad := \quad \sum_{m=1}^{N_a} \mathrm{softmax}\big(\mathbf{H}^{l-1} Q_m^l K_m^l (\mathbf{H}^{l-1})^T\big) \mathbf{H}^{l-1} V_m^l O_m^l =
\begin{bmatrix}
A_1^T(\mathbf{H}^{l-1}) \\
A_2^T(\mathbf{H}^{l-1}) \\
\vdots \\
A_N^T(\mathbf{H}^{l-1})
\end{bmatrix}
\in \mathbb{R}^{N_p \times D},
$$

where

$$
A_i(\mathbf{H}^{l-1}) = \sum_{m=1}^{N_a} [\sum_{j=1}^{N_p} M_{ij} (O_m^l)^T (V_m^l)^T (\mathbf{H}_j^{l-1})^T]
$$

and

$$
M_{i:} = \mathrm{softmax}(\mathbf{H}_i^{l-1} Q_m^l K_m^l (\mathbf{H}^{l-1})^T).
$$

For any $l = 1, ..., L$, by taking partial derivatives we obtain that

$$
\begin{aligned}
J_{ij}^l &= \sum_{m=1}^{N_a} [\sum_{t=1}^{N_p} M_{ij} (O_m^l)^T (V_m^l)^T] + \sum_{m=1}^{N_a} [\sum_{t=1}^{N_p} \mathbf{H}_j^{l-1} V_m^l O_m^l \frac{\partial M_{ij}}{\partial \mathbf{H}_t^{l-1}}] \\
&= \sum_{m=1}^{N_a} \sum_{t=1}^{N_p} \Big[ M_{it} (O_m^l)^T (V_m^l)^T + \mathbf{H}_t^{l-1} V_m^l O_m^l \mathbf{M}^i (E_{ti} \mathbf{H}^{l-1} Q_m^l K_m^l + \mathbf{H}^{l-1} (K_m^l)^T (Q_m^l)^T \delta_{ij}) \Big],
\end{aligned}
$$

where $\mathbf{M}^i := \text{diag}(M_{i:}) - M_{i:}^T M_{i:}$ with $\sup \|\mathbf{M}^i\| \leq \frac{1}{2}$, and $E_{ij} \in \mathbb{R}^{N_p \times N_p}$ is a binary matrix with zeros everywhere except the $(i,j)$-th entry.

Under assumption 1, for any $l = 1, ..., L$, we then have

$$\sup \|J_{ij}^l\| \leq C_{Lip} N_a B_O B_V B_K B_Q \|\mathbf{H}^{l-1}\|_2,$$

and

$$\sup \|J_{\mathbf{H}^{l-1}}[\mathcal{A}^l]\| \leq N_p \sup \|J_{ij}^l\| \leq C_{Lip} N_p N_a B_O B_V B_K B_Q \|\mathbf{H}^{l-1}\|_2,$$

where $C_{Lip}$ is a positive constant.

Then the Lipschitz constant of $L$ layer Transformer is

$$L_{\mathcal{T}} = C_{Lip} N_p^2 D B_K B_Q B_P \prod_{l=1}^{L} N_a^l B_{W_1}^l B_{W_2}^l B_O^l B_V^l. \tag{11}$$

Specifically, if the output Transformer model is assumed to take the last token at $L$-layer, its Lipschitz constant is

$$L_{\mathcal{T}}^* = N_p^{-\frac{1}{2}} L_{\mathcal{T}}. \tag{12}$$

This completes the proof.

## I NUMERICAL EVALUATIONS

**Evaluation on non-i.i.d data scenario:** In the non-i.i.d scenario, besides the asymptotic behavior and error accumulation, we additionally validate the impact of distributional alignment (quantified by $\|\mathbf{q} - \mathbf{v}\|$) in Corollary 4. We consider a complex scenario where the training and test tasks are drawn from related but non-identical distributions. Specifically, for each sample $i$, given a parameter vector $\beta^i$, we similarly generate a length-$L$ sequence via the recurrence relation $c_l^i = \beta_{l-1}^i c_{l-1}^i + \epsilon$ for $l = 1, \ldots, L$, where $c_0^i \sim \mathcal{N}(0, \mathbf{I}_d)$ is the query and $\epsilon \sim \mathcal{N}(0, 0.1 \cdot \mathbf{I}_d)$. Different from i.i.d scenario, in this scenario, the training parameter $\beta$ is drawn from a mixture of two Gaussian distributions $p_{\text{train}}(x) = p_1 \cdot \mathcal{N}(1, 0.1 \cdot \mathbf{I}_d) + p_2 \cdot \mathcal{N}(0.2, 0.1 \cdot \mathbf{I}_d)$, where $p_1$ and $p_2$ are the weights such that $p_1 + p_2 = 1$. The test data is drawn from a different distribution: $p_{\text{test}}(x) = \mathcal{N}(0.1, 0.1 \cdot \mathbf{I}_d)$. We consider the prompting format that incorporates a single in-context example. For training samples, the in-context example is drawn from the same distribution as the query task, i.e., both query and support samples share the same $\beta^i$. For test samples, the in-context example is instead drawn from $\mathcal{N}(0.2, 0.1 \cdot \mathbf{I}_d)$, regardless of the test query's distribution. This design aims to reduce the distributional divergence between the support and query examples in the test setting.

Under this setting, we assign group-wise importance weights based on the product of pairwise overlaps between the training and test component distributions:

$$G^{(k)} \propto \text{Overlap}(\mathcal{N}(\mu_k, \sigma_k), \mathcal{N}(0.1, 0.1 \cdot \mathbf{I}_d)) \cdot \text{Overlap}(\mathcal{N}(\mu_k, \sigma_k), \mathcal{N}(0.2, 0.1 \cdot \mathbf{I}_d)), k = 1, 2$$

where $k = 1, 2$ means the two class distributions of prompts in training process, and the total overlap between two distributions $p(x)$ and $q(x)$ is defined as $\text{Overlap}(p, q) = \int_{-\infty}^{\infty} \min\{p(x), q(x)\} \, dx$.

This value lies in the interval $[0, 1]$, where 1 indicates complete distributional alignment and 0 denotes no overlap. We then can approximate the optimal weights $\mathbf{v} \in \mathbb{R}^N$ as:

$$\mathbf{v} = (\underbrace{\frac{\bar{G}^{(1)}}{p_1 N}, \cdots, \frac{\bar{G}^{(1)}}{p_1 N}}_{p_1 N}, \underbrace{\frac{\bar{G}^{(2)}}{p_2 N}, \cdots, \frac{\bar{G}^{(2)}}{p_2 N}}_{p_2 N}) \in \mathbb{R}^N,$$

where $\bar{G}^{(k)}$ is re-nomalized constant such that $\sum_k \bar{G}^{(k)} = 1$, $p_1$ and $p_2$ are the proportions of the two training components, and $N$ is the total number of training samples. These weights are then uniformly assigned to all training prompts according to their source component. For training weights, we set $w = \{0, 0.2, 0.4, 0.6, 0.8, 1\}$ and let the training weights be

$$\mathbf{q} = (\underbrace{\frac{w}{p_1 N}, \cdots, \frac{w}{p_1 N}}_{p_1 N}, \underbrace{\frac{(1-w)}{p_2 N} \cdots, \frac{(1-w)}{p_2 N}}_{p_2 N}).$$

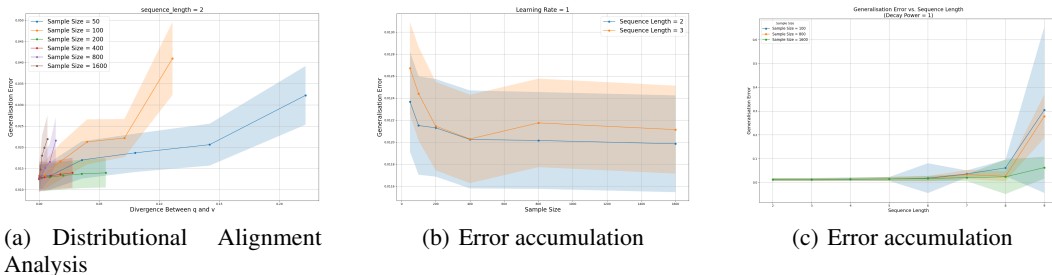

(a) Distributional Alignment Analysis

(b) Error accumulation

(c) Error accumulation

Figure 4: The generalisation error under non-i.i.d scenario.

*The impact of distributional alignment on non-i.i.d generalisation:* We evaluate the role of distribution mismatch by explicitly controlling the norm $\|\mathbf{q} - \mathbf{v}\|$, which quantifies the divergence between the empirical training distribution $\mathbf{q}$ and the ideal importance-weighted distribution $\mathbf{v}$. Figure 4(a) shows that the $\ell_2$ distance between $\mathbf{q}$ and $\mathbf{v}$ steadily increases, and the non-i.i.d. generalisation ability correspondingly deteriorates, manifesting as a larger generalisation error. This observation validates Corollary 4, which asserts that tighter alignment between the training and test prompt distributions yields better generalisation under distribution shift. Moreover, it underscores the importance of high-quality prompts for non-i.i.d. settings, since they reduce the gap between training and test distributions and thus improve generalisation.

*The Generalisation Error Convergence Analysis:* We evaluate the generalization error as the number of training samples $N$ increases. Figure 4(b) demonstrates that the error decreases and asymptotically vanishes as $N \to \infty$, consistent with the theoretical prediction in Corollary 4 for the non-i.i.d. setting.

*The Error Accumulation Analysis:* Figure 4(c) shows that the generalization error increases with sequence length, following an approximately logarithmic trend. In particular, once the sequence length exceeds a threshold near $\ln N$, the error rises sharply. Moreover, this threshold shifts to larger values as the sample size increases. These empirical findings support Theorem 5 under non-i.i.d.

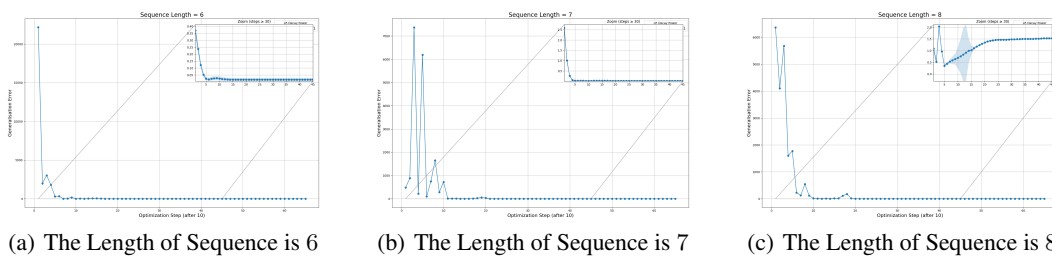

(a) The Length of Sequence is 6

(b) The Length of Sequence is 7

(c) The Length of Sequence is 8

Figure 5: Generalisation error progression over optimization steps in the non-i.i.d. setting.

*Overfitting Risk:* Figure 5 depicts how the generalisation error evolves with the number of optimization steps. As the sequence length increases, the task becomes more complex and the loss landscape grows more non-smooth, resulting in a heightened risk of overfitting. These observations align with the conclusions of Theorem 4.

*Empirical evaluation on realistic data:* We conduct an additional NLP experiment on a sentiment-classification task. The training set consists of labeled movie reviews, and the test-time prompts contain several review–label demonstration pairs. We collected approximately 600 movie reviews from Douban, segmented them into sentences, and fine-tuned a base GPT-2 model. Another 100 reviews were prepared for in-domain testing. Using the same procedure, we also constructed a literary-text test set from online literature platforms to create a distinct out-of-domain distribution.

After fine-tuning on movie reviews, we examined how the discrepancy measure $disc(\mathbf{q})$ relates to generalization behavior. To this end, we formed target-prompt mixtures spanning both movie-review and literary domains, with mixture ratios ranging from 0:7 to 7:0. The results are reported in Table 4. Mixtures containing a higher proportion of literary prompts correspond to larger $disc(\mathbf{q})$, as literary

texts differ more substantially from movie reviews (empirically, their bidirectional KL divergences are around 12). These higher-discrepancy mixtures exhibit moderately increased predictive loss, whereas mixtures more aligned with the training distribution (smaller $disc(\mathbf{q})$) show lower loss and stronger in-context performance. Overall, the observed trend is consistent with the qualitative dependence predicted by our theoretical analysis.

| Prompt Config | Loss | Top-1 Acc. | Prompt Config | Loss | Top-1 Acc. |
|---|---|---|---|---|---|
| 7:0 | 0.9319 | 90.78% | 3:4 | 0.9416 | 90.64% |
| 6:1 | 0.9363 | 90.71% | 2:5 | 0.9434 | 90.61% |
| 5:2 | 0.9374 | 90.66% | 1:6 | 0.9474 | 90.57% |
| 4:3 | 0.9398 | 90.64% | 0:7 | 0.9477 | 90.60% |

Table 4: Prompt configuration vs. performance on sentiment classification.

*The Validation of Assumption on Lipschitz Constant:* Although the theoretical and empirical constants need not coincide numerically, observing that the empirical estimates follow the same scaling laws across model sizes and datasets confirms the asymptotic tightness of our Lipschitz and smoothness bounds. To validate it, we approximate the constant by sampling multiple inputs, computing gradient and Hessian norms, and taking the maximum observed value. This approach effectively captures the dominant scaling behavior and serves as a reliable empirical proxy. As shown in Eqs. 7–9, the Lipschitz (smoothness) upper bound depends on factors such as QKV matrix size, model depth, and other architectural parameters. We varied these factors to examine their influence, with the results summarized in Table 5. The empirical results reveal clear scaling patterns of the Lipschitz constant with respect to key architectural parameters such as model depth, embedding dimensions and attention head. The consistent asymptotic behavior provides empirical evidence supporting the effectiveness of our theoretical Lipschitz (smoothness) bound.

| Number of Layers | # Lipschitz (Layers) | Attention Heads | Lipschitz (Heads) | Embedding Dim | Lipschitz (Embedding) |
|---|---|---|---|---|---|
| 12 | 19.38 | 4 | 19.38 | 1218 | 19.38 |
| 24 | 43.46 | 8 | 29.20 | 2506 | 48.52 |
| 36 | 59.52 | 32 | 38.71 | 5712 | 108.31 |
| 48 | 70.22 | – | – | 11044 | 908.74 |
| 60 | 820.52 | – | – | – | – |
| 72 | 938.16 | – | – | – | – |
| 84 | 1168.58 | – | – | – | – |
| 96 | 1961.24 | – | – | – | – |

Table 5: Lipschitz-related quantities across Transformer configurations.

## J THE USE OF LARGE LANGUAGE MODELS (LLMS)

We used LLMs (e.g., ChatGPT) only as a general-purpose writing assistant. Its roles were limited to polishing language (grammar and clarity), and concise rephrasing or shortening of paragraphs without adding technical content. The LLMs did not generate research ideas, problem formulations, proofs, theorems, algorithms, experiments, results, figures, or evaluations. All technical content (definitions, lemmas/theorems, proofs, algorithms), experimental designs, and conclusions are solely by the authors and were fully verified by us. The authors take full responsibility for all text in this paper. The LLM is not an author.

