# OpenReview forum: "Towards a Theoretical Understanding of In-context Learning: Stability and Non-I.I.D Generalisation"
_ICLR.cc/2026/Conference — ICLR 2026 Poster_

### Official Review · Reviewer_rUZ4 · 2025-10-26

**Soundness:** 3
**Presentation:** 3
**Contribution:** 2
**Rating:** 4
**Confidence:** 3

**Summary:**

This paper studies the ICL generalization of Transformers by characterizing the algorithmic stability and distributional discrepancy. The authors especially establish comprehensive discussion in different scenarios, including smooth or non-smooth loss functions, and i.i.d. or non-i.i.d. data. Some experiments are conducted for supporting the theory.

**Strengths:**

1. The theoretical analysis is impressive and looks solid.

2. The discussion is comprehensive enough to cover many cases.

**Weaknesses:**

1. The practical insight of the analysis is unclear. I am not sure how the proposed results can be used to explain any phenomenon or improve performance. This makes this work less interesting to me.

2. This paper mainly focuses on Transformers with multiple heads and multiple layers. However, it is not clear how the number of heads and layers affects the theoretical results. I don't know whether the derived bounds are tight and can be quantitatively verified by experiments. It is also not clear how and why the results and the analysis of Transformers differ from non-Transformer models. Therefore, I cannot specifically evaluate the novelty of the analysis.

3. The message from Section 4.1 is quite awkward. It combines both a brief introduction of the theoretical results and the main proof technique. However, a proof sketch should introduce the logic chain of establishing the proof rather than only mentioning the theoretical tools used in the proof.

**Questions:**

1. I don't quite get the discussion of Figure 2 (b). I cannot see why the generalization error increases following a "logarithmic" trend with sequence length.

2. It seems experiments in Section I are more interesting to justify the theoretical results. Why not put them in the main body?

---

> ### Author Response · Authors · 2025-11-21
> **Rebuttal to Reviewer rUZ4  (1/3)**
>
> Dear Reviewer rUZ4,
> Thank you for your positive support for our work. We appreciate your constructive suggestions and valuable questions. We now address your concerns in detail. Due to the character limit for each response, we slightly rearranged the order of the questions to ensure that our answers remain complete. Also, we have re-uploaded the revised PDF, and all changes have been highlighted in blue..
>
> **Q1. The practical insight of the analysis is unclear. I am not sure how the proposed results can be used to explain any phenomenon or improve performance.**
>
> A1. Thank you for raising this concern. We totally agree that the practical implications of our bounds need to be made more explicit, and we will revise the manuscript accordingly. Indeed, our theoretical results not only justify several empirical behaviors previously observed in practice, but also provide new guidance for model training and prompt selection.
>
> **First, our theory formalizes some well-known empirical observations.** Many empirical studies have shown that ICL performance is highly sensitive to how well the prompt distribution aligns with the model’s training distribution. For example, Raventós et al. identify a diversity threshold below which ICL fails to emerge, demonstrating that Transformer-based LLMs cannot perform new tasks through ICL when the pre-training distribution lacks sufficient diversity. Shin et al. further show that ICL only appears after combining two corpora, even though neither corpus alone yields ICL. In addition, pretraining on an in-domain corpus can improve ICL. These empirical phenomena are theoretically captured by our discrepancy measure $disc(\mathbf q)$ (Definition 6) and Theorems 1 and 3, which show that when the demonstrations  are aligned with the model’s pretraining distribution, the discrepancy $\mathrm{disc}(\mathbf q)$ remains small, leading to a tighter generalization bound.  In addition, we have included supplementary NLP experiments (see Appendix I). Here, the training set consists of labeled movie reviews, while test-time prompts contain several review–label demonstration pairs. We collected around 600 movie reviews from Douban, segmented them into sentences, and fine-tuned a base GPT-2 model. Another 100 reviews were prepared for testing. Using the same procedure, we also gathered a literary-text test set from online literature platforms. After fine-tuning on movie reviews, we evaluated how the discrepancy measure $disc(\mathbf q)$ influences generalization. We constructed target-prompt mixtures spanning both movie-review and literary domains, ranging from 0:7 to 7:0. The results are shown in the table below. A higher proportion of literary prompts implies a larger $disc(\mathbf q)$ because literary texts differ more substantially from movie reviews (bidirectional KL divergences ≈12), which in turn leads to higher predictive loss.
>
> These experimental results support the validity of our Thm. 4 in the non-i.i.d. setting, demonstrating that $\mathrm{disc}(q)$ indeed influences the model’s generalization performance. Due to time constraints, we plan to strengthen the empirical section with larger models and datasets in the final version of the paper.
>
> | Prompt Config |  Loss  | Top-1 Acc. |
> |---------------|-----------|-----------|
> | 7:0       | 0.9319 | 90.78%    |
> | 6:1       | 0.9363 | 90.71%    |
> | 5:2       | 0.9374 | 90.66%    |
> | 4:3       | 0.9398 | 90.64%    |
> | 3:4       | 0.9416 | 90.64%    |
> | 2:5       | 0.9434| 90.61%    |
> | 1:6       | 0.9474 | 90.57%    |
> | 0:7       | 0.9477 | 90.6%    |
>
> **Beyond the interpretability for empirical findings, our theory also provides new guidance for model training.** According to our generalization bound (Thm. 1), the discrepancy $\mathrm{disc}(\mathbf q)$ can be viewed as a function of the training weight vector $\mathbf q$. Although $\mathrm{disc}(\mathbf q)$ is defined as an expectation, prior work (e.g., [1]) shows that it can be accurately approximated using only training data. This enables a practical implementation, since we can learn a weighting function $\hat q$ that upweights samples that match the target query and enhance generalization.
>
> Ref
>
> [1] Vitaly Kuznetsov and Mehryar Mohri. Learning Theory and Algorithms for Forecasting Non-Stationary Time Series. NIPS, 2015

---

> ### Author Response · Authors · 2025-11-21
> **Rebuttal to Reviewer rUZ4 (2/3)**
>
> **Q2. It is not clear how the number of heads and layers affects the theoretical results, and It is also not clear how and why the results and the analysis of Transformers differ from non-Transformer models.**
>
> A2. We appreciate this question and agree that the dependence on Transformer architecture should be made more explicit. Our  results are indeed tailored to multi-head, multi-layer Transformer architectures in the ICL setting, and this dependence appears in several concrete ways:
>
> **(1) Architecture-dependent constants tied explicitly to Transformer structure.** In the original version, for brevity, we treated several architecture-dependent quantities (e.g., network depth, number of attention heads, layer widths) as constants and did not write them explicitly in the asymptotic expressions. More concretely, the gradient norms, Hessian norms, and Lipschitz (smoothness) constants in our analysis all depend explicitly on the Transformer depth $L$ and the number of attention heads $N_a$; see Appendix H for the detailed formulas. For example, the loss bound $M_\ell$ satisfies $M_\ell \sim (B_{W_1} B_{W_2} B_V B_O N_a)^{2L}$ and both $L_\ell$ and $\gamma$ accumulate layer-wise contributions. As a result, in Theorem 1 the stability coefficient $\beta$ depends on $M_\ell$, $L_\ell$, and $\gamma$, all of which grow with the Transformer depth L and the attention configuration. If we no longer treat L as fixed but allow it to scale with the sample size N, maintaining stability requires that the growth of L does not dominate the decay. Since quantities such as $M_\ell$ may grow exponentially in $L$, a sufficient condition for stability is that $L$ grows at most logarithmically with $N$, i.e., $L = O(\log N)$, under the vanilla SGD-style optimization considered in our analysis.
>
> **(2) Error accumulation is specific to autoregressive Transformer-style ICL.** Our error-accumulation analysis (e.g., Theorem 5) is not a generic neural-network result, but is derived for autoregressive next-token prediction, which is the standard way Transformers perform ICL. The bound tracks how prediction error compounds across the output token sequence, exploiting the causal, left-to-right structure of Transformer decoding. This accumulation behavior and the way it scales with the number of generated tokens is specific to the autoregressive Transformer setting, rather than to arbitrary feed-forward or encoder-only architectures.
>
> **Q3. Section 4.1 is awkward; a proof sketch should explain the logic chain, not just list tools.**
>
> A3. Thank you for this helpful comment. We restructure this section (Please see line 248-273) and hope these changes will make Section 4.1 much more readable and aligned with the reviewer’s expectation of a proof sketch.
>
> **Q4. I don’t quite get the discussion of Fig. 2(b). Why “logarithmic” trend with sequence length?**
>
> A4. Thank you for pointing this out. We acknowledge that the term “logarithmic” in the description of Fig. 2(b) was a typo. The correct trend should be polynomial, which is fully consistent with Theorem 5 (see Section G), where we show that the accumulated error grows polynomially with the prompt length. We have corrected this in the revised manuscript.
>
>
> **Q5. Experiments in Section I seem more interesting. Why not put them in the main body?**
>
> A5. We appreciate this suggestion and agree that the experiments in Section I provide strong empirical support for our theoretical claims. Initially, we placed them in the Appendix due to space constraints and to keep the main text focused on the theoretical development. However, we agree that moving (or summarizing) these experiments into the main body would substantially improve the paper. In the final version, we will promote the most informative experiments to the main text.

---

> ### Author Response · Authors · 2025-11-21
> **Rebuttal to Reviewer rUZ4 (3/3)**
>
> **Q3. I don't know whether the derived bounds are tight and can be quantitatively verified by experiments.**
>
> A3. We have incorporated two additional empirical evaluation on the effectiveness of the Lipschitz-related assumptions used in our analysis to further support the validity of our theoretical findings.  As with most theory work [1-3], we focus on provable upper bounds for the Lipschitz and smoothness constants (Appendix H) to obtain a generalization bound that holds for all inputs and distributions. Nevertheless, we agree that practical estimation is crucial, as it offers empirical evidence of how well the theoretical analysis reflects real-world behavior. Although the theoretical and empirical constants need not coincide numerically, observing that the empirical estimates follow the same scaling laws across model sizes and datasets confirms the asymptotic tightness of our Lipschitz and smoothness bounds.
>
> To validate it, we approximate the constant by sampling multiple inputs, computing gradient and Hessian norms, and taking the maximum observed value. This approach effectively captures the dominant scaling behavior and serves as a reliable empirical proxy. As shown in Eqs. 6–7, the Lipschitz (smoothness) upper bound depends on factors such as $QKV$ matrix size, model depth, and other architectural parameters. We varied these factors to examine their influence, with the results summarized in the following charts.
>
> | Number Layers | Lipschitz (Layers) | Attention Heads | Lipschitz (Heads) | Embedding Dim | Lipschitz (Embedding) |
> |---------------|--------------------|------------------|--------------------|----------------|-------------------------|
> | 12            | 19.38              | 4                | 19.38              | 128            | 19.38                   |
> | 24            | 43.45              | 8                | 29.50              | 256            | 48.92                   |
> | 36            | 50.52              | 32               | 39.71              | 512            | 108.31                  |
> | 48            | 70.72              | —                | —                  | 1024           | 968.74                  |
> | 60            | 820.25             | —                | —                  | —              | —                       |
> | 72            | 938.16             | —                | —                  | —              | —                       |
> | 84            | 1166.58            | —                | —                  | —              | —                       |
> | 96            | 1981.84            | —                | —                  | —              | —                       |
>
> The empirical results reveal clear scaling patterns of the Lipschitz constant with respect to key architectural parameters such as model depth, embedding dimensions and attention head. The consistent asymptotic behavior provides empirical evidence supporting the effectiveness of our theoretical Lipschitz (smoothness) bound.
>
> Moreover, standard regularization techniques commonly used in practice help ensure that these assumptions are effectively met. From the following Table, we observe that applying embedding, residual, or attention dropout significantly reduces the practical Lipschitz (smoothness) constant. Together with built-in mechanisms such as layer normalization that cap activation growth, these practices keep the model in a well-behaved regime, closely aligning practical behaviour with our theoretical assumptions.
>
> | Dropout Type        | Lipschitz Constant |
> |---------------------|--------------------|
> | No Dropout          | 19.38              |
> | Residual Dropout    | 3.17               |
> | Embedding Dropout   | 5.28               |
> | Attention Dropout   | 3.50               |
>
> We have added additional content in the revised version to further justify the reasonableness and tightness of our theoretical assumptions (please see lines 1726–1756).
>
> Ref
>
> [1] Yunwen Lei Antoine Ledent and Marius Kloft. Sharper Generalization Bounds for Pairwise Learning, NeurIPS 2020.
>
> [2]Yu Bai et al. Transformers as statisticians: Provable in-context learning with in-context algorithm selection. NeurIPS, 2024.
>
> [3]Yufeng Zhang et al. What and how does in-context learning learn? bayesian model averaging, parameterization, and generalization. AISTATS, 2025

---

### Official Review · Reviewer_bLpF · 2025-10-28

**Soundness:** 3
**Presentation:** 1
**Contribution:** 2
**Rating:** 4
**Confidence:** 3

**Summary:**

This paper derives generalization guarantees for non-linear multi-layer/multi-head Transformers under ICL, by coupling mini-batch-GD-dependent uniform stability with a hypothesis-space-independent discrepancy measure; the bounds highlight (i) optimization- and smoothness-aware choices of step-size/batch/iterations, (ii) the need to align prompt distributions between training and inference, and (iii) error accumulation across generated tokens implying an at-most logarithmic growth of prediction length for reliable generalization, corroborated by experiments.

**Strengths:**

The theoretical results are seemingly sound and cover both smooth and non-smooth regimes.

**Weaknesses:**

* The definitions of $\zeta_1$ and $ \zeta_2 $ are missing or unclear around lines 295–305. Please specify them explicitly for completeness.

* It is counter-intuitive that the non-smooth counterpart allows ( Q ) to be exponentially smaller than that in the smooth case (Corollary 2 vs. Corollary 1). The theoretical intuition behind this discrepancy should be clarified.

* The claim in lines 300–302 remains vague without quantitative support. A more formal analysis is needed to characterize the continuous Pareto frontier of the purported trade-off.

* The manuscript does not clearly explain why small-batch SGD achieves better generalization than its large-batch counterpart. A brief theoretical sketch in the proposed framework would strengthen the argument. Similarly, the Remarks section could more clearly outline how the asymptotic behaviors arise from the assumed settings, rather than only restating theorems.

* (Relatively minor point) Experiments on realistic datasets would make the findings more convincing and demonstrate the applicability of the theory.

**Questions:**

It would be good to include the discussions of arXiv:2508.09820 and arXiv:2411.02199, which consider generalization analysis over non-orthogonal data.

---

> ### Author Response · Authors · 2025-11-21
> **Rebuttal to Reviewer bLpf (1/3)**
>
> Dear Reviewer bLpF,
>
> Thank you for your careful evaluation of our work. We appreciate your constructive suggestions and valuable questions. We now address your concerns in detail. Due to the character limit for each response, we slightly rearranged the order of the questions to ensure that our answers remain complete; Also, we have re-uploaded the revised PDF, and all changes have been highlighted in blue.
>
> **Q1: The definitions of $\zeta_1$ and $\zeta_2$  are missing or unclear around lines 295–305. Please specify them explicitly for completeness.**
>
> A1. Thank you for pointing this out. We agree that the definitions in lines 295–305 were not sufficiently clear. In fact, $\zeta_1$ and $\zeta_2$ are arbitrary non-negative real numbers that control the growth rates of the batch size and iteration count, respectively. In the revision, we will explicitly define all relevant symbols at first use to ensure completeness and clarity (**please see line 326-327**).
>
> **Q2. Why the non-smooth case allows smaller Q than the smooth case**
>
> A2. Thanks for your question. We agree that, at first sight, it may seem counter-intuitive that the non-smooth counterpart allows Q to be exponentially smaller than in the smooth case. This discrepancy, however, stems from the fundamentally different mechanisms that control stability in the two regimes.
>
> In the non-smooth setting, SGD updates are non-expansive but not contractive: the loss is only Lipschitz, and there is no smoothness to damp perturbations. As a consequence, the distance between two neighboring trajectories may be preserved rather than shrunk at each step, and the stability term accumulates essentially linearly with the number of updates, yielding O(Q/N)-type bounds. To ensure that this stability term still vanishes with N, Q must grow sufficiently slowly so that the 1/N decay dominates; a standard sufficient condition in this regime is Q = O(log N). This logarithmic restriction is a standard condition in algorithmic stability analysis, e.g., [1].
>
> Ref
>
> [1] Sejun Park et al. Generalization Bounds for Stochastic Gradient Descent via Localized ε-Covers. NeurIPS, 2022.
>
> **Q3. The claim in lines 300–302 remains vague without quantitative support. A more formal analysis is needed to characterize the continuous Pareto frontier of the purported trade-off.**
>
> A3.  We thank the reviewer for this comment and agree that our discussion in lines 300–302 was too qualitative. We clarify the intended trade-off and how it can be formalized within our framework.
> Corollary 1 captures a fundamental trade-off between optimization and stability. Increasing the number of iterations $Q$ (and/or the batch size) generally improves optimization and reduces the empirical risk (which is observable). At the same time, our stability analysis shows that larger $Q$ amplifies the accumulated perturbations along the optimization path, thereby worsening the stability coefficient $\beta$ and enlarging the generalization gap. Importantly, this behavior of stability is a standard phenomenon in the algorithmic-stability literature (e.g.,  [1,2]), where stability typically deteriorates as the number of update steps grows.
> Formally, under the parameterization $|B|=O(N^{\zeta_1})$, $Q=O(N^{\zeta_2})$, Corollary 1 yields a stability term of order $\beta=O(𝑁^{\zeta_1+\zeta_2\gamma/(1+\gamma)-1})$ and the condition $\zeta_1+\zeta_2\gamma/(1+\gamma)<1$ defines the feasible region of stable training schemes. Within this region, larger $\zeta_2$ (more iterations) typically leads to lower empirical risk but higher $\beta$, while smaller $\zeta_2$ improves stability at the cost of less optimization. The boundary of the feasible region corresponds to a Pareto frontier in this induced space: moving closer to perfect optimization (more iterations/bigger batches) inevitably pushes us toward worse stability, and vice versa.
>
> In the revision, following your constructive suggestion, we will make this optimization–stability trade-off explicit (please see line 315-318).
>
> Ref
>
> [1] Hardt, M., Recht, B., and Singer, Y. Train faster, generalize better: Stability of stochastic gradient descent. ICML, 2016.
>
> [2] Sejun Park et al. Generalization Bounds for Stochastic Gradient Descent via Localized ε-Covers, NeurIPS, 2022.

---

> ### Author Response · Authors · 2025-11-21
> **Rebuttal to Reviewer bLpf (2/3)**
>
> **Q4. Why small-batch SGD generalizes better than large-batch SGD**
>
> A4. Thank you for pointing this out.  From our upper bound, the stability coefficient scales as $\beta \propto B$, so smaller batches yield a tighter generalization bound. This behavior is also intuitively reasonable: the variance of the mini-batch gradient increases as the batch size decreases, injecting stochastic noise into the updates. Such noise is widely understood to provide implicit regularization, effectively smoothing the optimization trajectory and reducing the functional Lipschitz constant. In contrast, large batches suppress this stochasticity, weakening the implicit regularization effect and leading to a larger stability coefficient $\beta$.
>
> Additionally, following your constructive suggestion, the Remarks will be revised to clarify how the asymptotic behaviors (both in the smooth and non-smooth regimes) arise directly from the assumptions, rather than restating the theorems. **For clarity, we have reorganized the proof sketch in Section 4.1 to better highlight the logical flow of the argument. In addition, we now explain how the asymptotic behaviors arise in our results, including the revised discussions accompanying Corollaries 1 and 2.**
>
> **Q5. Discussion of arXiv:2508.09820 and arXiv:2411.02199**
>
> A6. We thank the reviewer for pointing us to these two highly insightful works [1,2]. These papers make significant contributions by revealing how structured, non-orthogonal concept geometries and task-vector mechanisms emerge inside pretrained Transformers, and by offering rigorous optimization-based explanations for phenomena such as multi-concept semantic composition, OOD generalization, and factual-recall vector arithmetic. Their analyses highlight elegant connections between latent geometric structure, attention dynamics, and the emergence of ICL.
>
> Our work focuses on a complementary aspect of the theory and we believe these perspectives are mutually reinforcing. Rather than assuming a particular latent concept geometry, we study distribution-shift–aware generalization through the lens of algorithmic stability. We introduce a discrepancy measure that quantifies how prompt distributions deviate from the training distribution and derive PAC-style bounds for multi-head, multi-layer Transformers trained via mini-batch SGD. Thus, our results address a different but compatible question: how ICL generalizes as the prompt distribution changes, independent of any assumed semantic geometry.
> In the revised manuscript, we added a dedicated discussion acknowledging the strengths of these two works and clarifying how our contributions address a complementary aspect of the problem (**please see line 108-115**).
>
> Ref
>
> [1] Dake Bu et al. Provably Transformers Harness Multi-Concept Word Semantics for Efficient In-Context Learning. NeurIPS, 2024.
>
> [2] Dake Bu et al. Provable In-Context Vector Arithmetic via Retrieving Task Concepts. ICML, 2025.

---

> ### Author Response · Authors · 2025-11-21
> **Rebuttal to Reviewer bLpf (3/3)**
>
> **Q5. Experiments on realistic datasets would make the findings more convincing and demonstrate the applicability of the theory**
>
> A5. Thank you for the suggestion. We have added two empirical studies to support our theory (lines 1701–1746): one validating the Lipschitz-related assumptions and another confirming our theorems on an NLP sentiment-classification task.
>
> **i) Practicality and justification of the boundedness and Lipschitz smoothness assumptions.**  As with most theory work  [1-3], we derive provable upper bounds for the Lipschitz and smoothness constants (Appendix H) so that the generalization guarantee holds for all inputs and distributions. We agree, however, that empirical estimation is important. The fact that the empirical constants follow the same scaling laws across model sizes and datasets supports the asymptotic tightness of our bounds.
>
> To validate it, we approximate the constant by sampling multiple inputs, computing gradient and Hessian norms, and taking the maximum observed value. This approach effectively captures the dominant scaling behavior and serves as a reliable empirical proxy. As shown in Eqs. 6–7, the Lipschitz (smoothness) upper bound depends on factors such as $QKV$ matrix size, model depth, and other architectural parameters. We varied these factors to examine their influence, with the results summarized in the following charts.
>
> | Number Layers | Lipschitz (Layers) | Attention Heads | Lipschitz (Heads) | Embedding Dim | Lipschitz (Embedding) |
> |---------------|--------------------|------------------|--------------------|----------------|-------------------------|
> | 12            | 19.38              | 4                | 19.38              | 128            | 19.38                   |
> | 24            | 43.45              | 8                | 29.50              | 256            | 48.92                   |
> | 36            | 50.52              | 32               | 39.71              | 512            | 108.31                  |
> | 48            | 70.72              | —                | —                  | 1024           | 968.74                  |
> | 60            | 820.25             | —                | —                  | —              | —                       |
> | 72            | 938.16             | —                | —                  | —              | —                       |
> | 84            | 1166.58            | —                | —                  | —              | —                       |
> | 96            | 1981.84            | —                | —                  | —              | —                       |
>
> The empirical results reveal clear scaling patterns of the Lipschitz constant with respect to key architectural parameters such as model depth, embedding dimensions and attention head. The consistent asymptotic behavior provides empirical evidence supporting the effectiveness of our theoretical Lipschitz (smoothness) bound.
>
> ii) We include an additional NLP experiment to demonstrate our generalization bounds in realistic in-context learning scenarios to make the theoretical insights more accessible to readers, where we consider a standard sentiment-classification task. The training set consists of labeled movie reviews, and the test prompts include several review–label demonstration pairs. We collected about 600 Douban movie reviews for fine-tuning GPT-2 and over 100 for testing, covering positive, neutral, and negative sentiments. We also gathered literary-text test data using the same segmentation procedure. After fine-tuning on the movie-review corpus, we examined how the discrepancy measure $disc(\mathbf q)$ affects generalization by evaluating mixtures of movie-review and literary prompts, ranging from 0:7 to 7:0. The results are shown in the table below. A higher proportion of literary prompts generally corresponds to a larger $disc(\mathbf q)$, as literary texts differ from movie reviews in both style and distribution (with bidirectional KL divergences of roughly 12). This increased discrepancy generally leads to higher predictive loss, whereas a smaller $\mathrm{disc}(\mathbf{q})$ corresponds to lower loss and thus reflects stronger in-context learning capability.
>
> | Prompt Config |  Loss  | Top-1 Acc. |
> |---------------|--------|----------------|
> | 7:0       | 0.9319 | 90.78%    |
> | 6:1       | 0.9363 | 90.71%    |
> | 5:2       | 0.9374 | 90.66%    |
> | 4:3       | 0.9398 | 90.64%    |
> | 3:4       | 0.9416 | 90.64%    |
> | 2:5       | 0.9434| 90.61%    |
> | 1:6       | 0.9474 | 90.57%    |
> | 0:7       | 0.9477 | 90.6%    |
>
>
> Ref
>
> [1] Yunwen Lei Antoine Ledent and Marius Kloft. Sharper Generalization Bounds for Pairwise Learning, NeurIPS 2020.
>
>  [2]Yu Bai et al. Transformers as statisticians: Provable in-context learning with in-context algorithm selection. NeurIPS, 2024.
>
> [3]Yufeng Zhang et al. What and how does in-context learning learn? bayesian model averaging, parameterization, and generalization. AISTATS, 2025

---

> ### Comment · Reviewer_bLpF · 2025-11-27
>
> Thanks you for your reply.
>
> > Q1
>
> It is not appropriate to use informal argument "be arbitrary non-negative real numbers that control the growth rates of the batch size and iteration count". The formal definition should be clearly defined in a formal corollary, unless its role is an "informal corollary".
>
> Also, it might be not standard to put proof sketch ahead. Usually the readers want to see the main theorems first, followed by the proof sketch IMO.
>
> Besides, for Theorem 1, could you listed the most recent existing studies's results (e.g., arXiv:2410.01405 etc) to demonstrate existing work's rate w.r.t smooth constant, iteration, etc? I have no idea whether your rate is good or not.
>
> >  Q2-4, regarding small Q, B required for your controlled beta
>
> Your explanations didn't convince me that these are good. You aim to clarify the difference of techniques of two regimes, but didn't explain your smooth case to me. It seems that it is due to your techniques such that the Q and B need to be small, in order to ensure the desired beta. I found both corollaries are unrealistic or counter-intuitive: you want an upper bound of \beta, but the batch size is upper-bounded by an unsuitably large order, usually a large batch size could ensure within-batch empirical sample distribution stable instead of your proposed stochastic noise; you require upper-bound of Q, this is un-realistic since usually optimization analyses prefer scenarios where iteration is larger than a threshold, unless you focus on the benefit of early-stopping.
>
> Also, in my intuition, when the Q or B is large enough, by the view of martingale differences of population GD (good generalization) vs SGD, the optimisation and generalization would become better and be more close to the trajectory of population GD (arXiv:2411.02199). However, you here focus on how the beta could be controlled in an early-stopping scenario, claiming that large Q amplifies error. I failed to get the reasons despite you listed old [1,2], please explain more clearly how this amplification error forms, and whether they are due to your techniques and condition realm (e.g., there could be analyses under the conditions of long-iteration SGD where the generalisation error is controlled by the law of large number, but currently not discussed). Also, your updated 313-316 didn't characterize the pareto frontier explicitly, and the readers might fail to grasp the trade-off in a second.
>
>
> > Q5
>
> It might be good if Table 1 could include the work.

---

### Official Review · Reviewer_gCrj · 2025-11-01

**Soundness:** 3
**Presentation:** 3
**Contribution:** 3
**Rating:** 6
**Confidence:** 4

**Summary:**

This paper develops a theoretical framework for understanding the generalisation behavior of in-context learning (ICL) in Transformers under non-i.i.d. settings. The authors derive a generalisation error bound for the algorithmic stability and distributional discrepancy measure and conduct empirical evaluations to validate their theoretical findings.

**Strengths:**

1. This paper studies an interesting and important question regarding the stability and non-i.i.d. generalization of in-context learning (ICL), and it may offer valuable practical insights.

2. According to Table 1, this work considers more general and realistic settings compared to prior theoretical studies.

3. The paper is well-presented, featuring clear illustrative figures, concise proof sketches, and well-organized comparisons with related works (e.g., Table 1).

**Weaknesses:**

1. The experiments are purely synthetic and do not include realistic NLP or multimodal datasets. This limits practical impact.

2. Boundedness and Lipschitz smoothness may not hold for real Transformer loss landscapes; discussion of how these assumptions approximate practice would help.

3. It seems that these assumptions (boundedness and Lipschitz smoothness) are very general and could apply to any neural network architectures or tasks. Therefore, it is unclear whether the theoretical results in this paper truly provide any insights that are specific to the Transformer architecture or the in-context learning (ICL) problem.

**Questions:**

1. Could the authors include more realistic ICL tasks in the experiments to better support and validate their theoretical claims?
2. Could the authors provide a more detailed discussion on the practicality and justification of the boundedness and Lipschitz smoothness assumptions in real-world settings?
3. These assumptions (boundedness and Lipschitz smoothness) appear to be quite general and could potentially apply to many neural network architectures or tasks. Do the theoretical results in this paper offer insights that are specific to the Transformer architecture or the ICL problem in particular?

---

> ### Author Response · Authors · 2025-11-21
> **Rebuttal to Reviewer gCrj (1/3)**
>
> Dear Reviewer gCrj,
>
> Thank you for your positive support and valuable suggestions. We now address your concerns in detail (We have re-uploaded the revised PDF, and all changes have been highlighted in blue).
>
> **Q1: Boundedness and Lipschitz smoothness may not hold for real Transformer loss landscapes.**
>
>  A1: We appreciate the reviewer’s insightful comment. As with most theory work [1][2][3], we focus on provable upper bounds for the Lipschitz and smoothness constants (please see Appendix H) to obtain a generalization bound that holds for all inputs and distributions. Nevertheless, we agree that practical estimation is crucial, as it offers empirical evidence of how well the theoretical analysis reflects real-world behavior. Theorefore, observing that the empirical estimates follow the same scaling laws across model sizes and datasets confirms the asymptotic tightness of our Lipschitz and smoothness bounds.
>
> To validate it, we approximate the constant by sampling multiple inputs, computing gradient and Hessian norms, and taking the maximum observed value. This approach effectively captures the dominant scaling behavior and serves as a reliable empirical proxy. As shown in Eqs. 6–7, the Lipschitz (smoothness) upper bound depends on factors such as QKV matrix size, model depth, and other architectural parameters. We varied these factors to examine their influence, with the results summarized in the following charts.
>
> | Number Layers | Lipschitz (Layers) | Attention Heads | Lipschitz (Heads) | Embedding Dim | Lipschitz (Embedding) |
> |---------------|--------------------|------------------|--------------------|----------------|-------------------------|
> | 12            | 19.38              | 4                | 19.38              | 128            | 19.38                   |
> | 24            | 43.45              | 8                | 29.50              | 256            | 48.92                   |
> | 36            | 50.52              | 32               | 39.71              | 512            | 108.31                  |
> | 48            | 70.72              | —                | —                  | 1024           | 968.74                  |
> | 60            | 820.25             | —                | —                  | —              | —                       |
> | 72            | 938.16             | —                | —                  | —              | —                       |
> | 84            | 1166.58            | —                | —                  | —              | —                       |
> | 96            | 1981.84            | —                | —                  | —              | —                       |
>
> The empirical results reveal clear scaling patterns of the Lipschitz constant with respect to key architectural parameters such as model depth, embedding dimensions and attention head. The consistent asymptotic behavior provides empirical evidence supporting the effectiveness of our theoretical Lipschitz (smoothness) bound.
>
> Moreover, standard regularization techniques commonly used in practice help ensure that these assumptions are effectively met. From the following Table, we observe that applying embedding, residual, or attention dropout significantly reduces the practical Lipschitz (smoothness) constant. Together with built-in mechanisms such as layer normalization that cap activation growth, these practices keep the model in a well-behaved regime, closely aligning practical behaviour with our theoretical assumptions.
>
> | Dropout Type        | Lipschitz Constant |
> |---------------------|--------------------|
> | No Dropout          | 19.38              |
> | Residual Dropout    | 3.17               |
> | Embedding Dropout   | 5.28               |
> | Attention Dropout   | 3.50          |
>
> Indeed, the boundedness assumptions in our theoretical analysis can be further relaxed to unbounded settings, with the theoretical results still holding. For example, one can replace the assumption of a hard bound on inputs or Lipschitz constant with a light-tailed distribution assumption (e.g., inputs or features have sub-Gaussian tails) [4]. This means extremely large input values are exponentially unlikely, effectively limiting the influence of outliers without requiring an absolute bound.
>
> In the revised manuscript, we include a dedicated paragraph discussing the effectiveness of Lipschitz assumptions (**please see line 1726-1746**) and the weaker assumption (**please see line 292-297**).
>
> Refs:
>
> [1] Yunwen Lei  Antoine Ledent and Marius Kloft. Sharper Generalization Bounds for Pairwise Learning, NeurIPS 2020.
>
> [2]Yu Bai et al. Transformers as statisticians: Provable in-context learning with in-context algorithm selection. NeurIPS, 2024.
>
> [3]Yufeng Zhang et al. What and how does in-context learning learn? bayesian model averaging, parameterization, and generalization. AISTATS, 2025.
>
> [4] Amit Attia and Tomer Koren. A General Reduction for High-Probability Analysis with General Light-Tailed Distributions, arXiv, 2025.

---

> ### Author Response · Authors · 2025-11-21
> **Rebuttal to Reviewer gCrj(2/3)**
>
> **Q2: Could the authors include more realistic ICL tasks in the experiments to better support and validate their theoretical claims?**
>
> A2. We appreciate this suggestion and agree that adding a realistic in-context learning task will strengthen the paper. We include an additional NLP experiment to demonstrate our generalization bounds in realistic in-context learning scenarios to make the theoretical insights more accessible to readers, where we consider a standard sentiment-classification task.
>
> The training dataset contains labeled movie reviews, while the test-time prompt provides several demonstration pairs, each consisting of a review and its sentiment label. We selected movie reviews from Douban and segmented them by sentence, obtaining approximately 600 movie review entries as training data for fine-tuning the base gpt-2 model. The content includes positive reviews, neutral reviews, and negative reviews. Using the same method, we obtained over 100 entries as test data. We also collected literary-text test data from online literature platforms using the same segmentation strategy. After fine-tuning the large language model on the movie-review corpus, we focused on examining how the discrepancy measure $disc(\mathbf q)$ affects the generalization error. Accordingly, the target-prompt configurations used for evaluation covered both movie-review and literary domains, with cross-domain mixtures ranging from 0:7 to 7:0. The test results are summarized in the table below. In cross-domain evaluation, a higher proportion of literary prompts generally corresponds to a larger $disc(\mathbf q)$, as literary texts differ more markedly from movie reviews in both style and distribution (with bidirectional KL divergences of roughly 12). This increased discrepancy generally leads to higher predictive loss, whereas a smaller $\mathrm{disc}(\mathbf{q})$ corresponds to lower loss and thus reflects stronger in-context learning capability.
>
> These experimental results are incorporated into revised manuscript  (**please see Appendix I; line 1702-1725**), and further support the validity of our Thm. 4 in the non-i.i.d. setting, demonstrating that $\mathrm{disc}(q)$ indeed influences the model’s generalization performance. Due to time constraints, we will strengthen the empirical section with larger models and datasets in the final manuscript.
>
> | Prompt Config |   Loss   | Top-1 Acc. |
> |---------------|----------|------------|
> | 7:0           | 0.9319   | 90.78%     |
> | 6:1           | 0.9363   | 90.71%     |
> | 5:2           | 0.9374   | 90.66%     |
> | 4:3           | 0.9398   | 90.64%     |
> | 3:4           | 0.9416   | 90.64%     |
> | 2:5           | 0.9434   | 90.61%     |
> | 1:6           | 0.9474   | 90.57%     |
> | 0:7           | 0.9477   | 90.60%     |

---

> ### Author Response · Authors · 2025-11-21
> **Rebuttal to Reviewer gCrj (3/3)**
>
> **Q3: Do the theoretical results in this paper offer insights that are specific to the Transformer architecture or the ICL problem in particular?**
>
> A3. Thank you for the question. We understand the concern and respectfully clarify that, although boundedness and Lipschitz smoothness are general assumptions, our results are indeed tailored to Transformers performing in-context learning (ICL) in several concrete ways.
>
> **(1) Architecture-dependent constants tied explicitly to Transformer structure.** In the original version, for brevity, we treated several architecture-dependent quantities (e.g., network depth, number of attention heads, layer widths) as constants and did not write them explicitly in the asymptotic expressions. More concretely, the gradient norms, Hessian norms, and Lipschitz (smoothness) constants in our analysis all depend explicitly on the Transformer depth $L$ and the number of attention heads $N_a$; see Appendix H for the detailed formulas. For example, the loss bound $M_\ell$ satisfies $M_\ell \sim (B_{W_1} B_{W_2} B_V B_O N_a)^{2L}$ and both $L_\ell$ and $\gamma$ accumulate layer-wise contributions. As a result, in Theorem 1 the stability coefficient $\beta$ depends on $M_\ell$, $L_\ell$, and $\gamma$, all of which grow with the Transformer depth L and the attention configuration. If we no longer treat L as fixed but allow it to scale with the sample size N, maintaining stability requires that the growth of $L$ does not dominate the decay. Since quantities such as $M_\ell$ may grow exponentially in $L$, a sufficient condition for stability is that $L$ grows at most logarithmically with $N$, i.e., $L = O(\log N)$, under the vanilla SGD-style optimization considered in our analysis.
>
> **(2) Error accumulation is specific to autoregressive Transformer-style ICL.** Our error-accumulation analysis (e.g., Theorem 5) is not a generic neural-network result, but is derived for autoregressive next-token prediction, which is the standard way Transformers perform ICL. The bound tracks how prediction error compounds across the output token sequence, exploiting the causal, left-to-right structure of Transformer decoding. This accumulation behavior and the way it scales with the number of generated tokens is specific to the autoregressive Transformer setting, rather than to arbitrary feed-forward or encoder-only architectures.
>
> **(3) ICL-specific insights: demonstrations, context length, and attention.** Several of our theoretical insights are also specific to the ICL usage of Transformers, rather than to generic supervised learning: We analyze how the number of demonstration tokens (i.e., the context length devoted to in-context examples) affects generalization in ICL, which directly reflects how the self-attention mechanism aggregates information from demonstrations and query tokens.
>
> We will revise the paper to emphasize these Transformer- and ICL-specific aspects more clearly by making the dependence on depth and number of heads explicit in the main theorems (**please see line 313-316**).

---

### Official Review · Reviewer_J7sq · 2025-11-03

**Soundness:** 3
**Presentation:** 3
**Contribution:** 2
**Rating:** 4
**Confidence:** 4

**Summary:**

In this paper, generalization of in-context learning is studied using the tools of algorithmic stability. First a stability bound for transformer based architecture is derived, and then such a stability bound is coupled with discrepancy measure and provides the generalization guarantees for i.i.d sequences and non i.i.d sequences. Empirical investigation on synthetic tasks are provided to support the theory.

**Strengths:**

The paper tackles an important question of generalization of language models and proposes a general framework and approach for generalization bound for incontext learning using algorithmic stability and discrepancy measure. The flexible approach can also handel non i.i.d data scenario's with discrepancy measure and is interesting.

**Weaknesses:**

A following things limit the applicability or significance of the result.

i) A key limitation of the paper is that its results lack clear interpretability, and the novelty of the proposed approach is not effectively communicated to the reader. The work would be significantly strengthened by presenting a concrete example, such as one involving in-context learning. This would provide a practical scenario where the derived generalization bounds are tangible and make intuitive sense, helping to ground the paper's theoretical contribution

ii) What is the specificity of the result to incontext learning or transformer architecture ?

iii) The exteme dependence on  the iteration number, the stability becomes worse and worse with the  number of iterations (Q) and it is sometimes logarithmic in the Q meaning the result is not applicable for a single pass over the data.

**Questions:**

i) In the context of Theorem 2, do you give an example of scenario when $\beta \| q\|_2 N \to 0$ ?

ii) In table 2, the paper presents the convergence rate, does it not take into account the convergence of training loss ?

---

> ### Author Response · Authors · 2025-11-21
> **Rebuttal to Reviewer J7sq (1/3)**
>
> Dear Reviewer J7sq,
>
> Thank you for your careful evaluation of our work. We appreciate your constructive suggestions and valuable questions. We now address your concerns in detail (We have re-uploaded the revised PDF, and all changes have been highlighted in blue.).
>
> **Q1: A key limitation of the paper is that its results lack clear interpretability, and the novelty of the proposed approach is not effectively communicated to the reader.**
>
> A1: We appreciate the reviewer’s suggestion and fully agree that our theoretical results would benefit from more intuitive explanations.
> Our theoretical results not only justify several empirical behaviors previously observed in practice, but also provide new guidance for model training.
>
> First, our theory formalizes some well-known empirical observations: Many empirical studies have shown that ICL performance is highly sensitive to how well the prompt distribution aligns with the model’s training distribution. For example, Raventós et al. identify a diversity threshold below which ICL fails to emerge, demonstrating that Transformer-based LLMs cannot perform new tasks through ICL when the pre-training distribution lacks sufficient diversity [1]. Shin et al. further show that ICL only appears after combining two corpora, even though neither corpus alone yields ICL. In addition, pretraining on an in-domain corpus can improve ICL [2]. These empirical phenomena are theoretically captured by our discrepancy measure $disc(\mathbf q)$ (Definition 6) and Theorems 1 and 3, which show that when the demonstrations  are aligned with the model’s pretraining distribution, the discrepancy $\mathrm{disc}(\mathbf q)$ remains small, leading to a tighter generalization bound.
>
> Beyond the interpretability for empirical findings, our theory also provides new guidance for model training. According to our generalization bound (Thm. 1), the discrepancy $\mathrm{disc}(\mathbf q)$ can be viewed as a function of the training weight vector $\mathbf q$. Although $\mathrm{disc}(\mathbf q)$ is defined as an expectation, prior work (e.g., [3]) shows that it can be accurately approximated using only training data. This enables a practical implementation, since we can learn a weighting function $\hat{\mathbf q}$ that upweights samples that match the target query and enhance generalization.
>
> To further enhance the interpretability of our theoretical results, in the revised manuscript, we have added i) an additional remark explaining how our framework can guide model training (**please see Remark 5; line 463-472**); ii) additional NLP experiments on a sentiment-classification task (**please see Appendix I; line 1702-1725**). Here, the training set consists of labeled movie reviews, while test-time prompts contain several review–label demonstration pairs. We collected around 600 movie reviews from Douban, segmented them into sentences, and fine-tuned a base GPT-2 model. Another 100 reviews were prepared for testing. Using the same procedure, we also gathered a literary-text test set from online literature platforms. After fine-tuning on movie reviews, we evaluated how the discrepancy measure $disc(\mathbf q)$ influences generalization. We constructed target-prompt mixtures spanning both movie-review and literary domains, ranging from 0:7 to 7:0. The results are shown in the table below. A higher proportion of literary prompts implies a larger $disc(\mathbf q)$ because literary texts differ more substantially from movie reviews (bidirectional KL divergences ≈12), which in turn leads to higher predictive loss. These experimental results further support the validity of our Thm. 4 in the non-i.i.d. setting, demonstrating that $\mathrm{disc}(\mathbf q)$ indeed influences the model’s generalization performance. Due to time constraints, we willstrengthen the empirical section with larger models and datasets in the final manuscript.
>
> | Prompt Config |   Loss   | Top-1 Acc. |
> |---------------|----------|------------|
> | 7:0           | 0.9319   | 90.78%     |
> | 6:1           | 0.9363   | 90.71%     |
> | 5:2           | 0.9374   | 90.66%     |
> | 4:3           | 0.9398   | 90.64%     |
> | 3:4           | 0.9416   | 90.64%     |
> | 2:5           | 0.9434   | 90.61%     |
> | 1:6           | 0.9474   | 90.57%     |
> | 0:7           | 0.9477   | 90.60%     |
>
> Refs
>
> [1] Allan Raventós, Mansheej Paul, F. Chen, and Surya Ganguli.  Pretraining task diversity and the emergence of non-bayesian in-context learning for regression.  NeurIPS, 2023.
>
> [2] Seongjin Shin, Sang-Woo Lee, Hwijeen Ahn, Sungdong Kim, Hyoungseok Kim, Boseop Kim, Kyunghyun Cho, Gichang Lee, Woo Chul Park, Jung-Woo Ha, and Nako Sung.  On the effect of pretraining corpora on in-context learning by a large-scale language model.  NAACL, 2022.
>
> [3] Vitaly Kuznetsov and Mehryar Mohri. Learning Theory and Algorithms for Forecasting Non-Stationary Time Series. NIPS, 2015.

---

> ### Author Response · Authors · 2025-11-21
> **Rebuttal to Reviewer  J7sq (2/3)**
>
> **Q2: What is the specificity of the result to incontext learning or transformer architecture ?**
>
> A2: Thanks for your question. Our theoretical results are indeed tailored to in-context learning on Transformer architectures. In the previous version of the manuscript, since our focus was mainly on asymptotic behavior, we treated several architecture-dependent parameters (such as network depth and the number of attention heads) as constants and omitted them from the analysis. In fact, these parameters can be incorporated into the generalization analysis.
>
> Specifically, the gradient norms, Hessian norms, and the Lipschitz (smooth) constants used in our analysis all depend explicitly on the Transformer depth $L$ and the number of attention heads $N_a$ (see Appendix H for the detailed expressions). Therefore, the theoretical guarantees in our stability bounds naturally extend to these architecture-dependent parameters. For example, in Theorem 1, the stability coefficient $\beta$ depends on $M_\ell$, $L_\ell$, and $\gamma$, all of which grow with the network depth $L$. Since quantities such as $M_\ell$ grow exponentially in $L$ (see Eq. (6)), a sufficient condition for stability is that the depth grows at most logarithmically with $N$. In other words, under the vanilla SGD-style optimization considered in our analysis, one can ensure stability by restricting $L = O(\log N)$. In the revised manuscript, we explicitly discuss the relation between our theoretical results and Transformer-related architectural parameters  (**please see line 313-316**).
>
> **Q3: The exteme dependence on the iteration number, the stability becomes worse and worse with the number of iterations (Q) and it is sometimes logarithmic in the Q meaning the result is not applicable for a single pass over the data.**
>
> A3:  We thank the reviewer for raising this point. We first clarify that the phrase “a single pass over the data’’ appears to refer, if we understand correctly, to a sample-wise SGD setting, where one epoch requires $Q=N$ updates. Our analysis, however, is stated for the mini-batch SGD regime used in Transformer training, in which a single pass requires only $Q=N/B$ updates. Thus, even a logarithmic dependence on $Q$ remains fully compatible with practical single-pass or few-pass training when the batch size is large.
> Regarding the logarithmic requirement in the non-smooth case, this is a standard sufficient condition in algorithmic stability analysis, e.g., [1]. Without smoothness, SGD updates are not contractive. Deviations between neighboring trajectories do not shrink, so the stability term necessarily accumulates with the number of updates. Requiring $Q=O(log N)$ is therefore the weakest condition that ensures a non-vacuous bound in the non-smooth setting. Importantly, this restriction disappears under smooth losses. As shown in Corollary 1, when the loss is smooth (an assumption commonly made in theoretical analyses of Transformers), the bound allows $Q$ to scale polynomially with $N$, covering both single-pass and multi-epoch training.
>
> Ref
>
> [1] Sejun Park et al. Generalization Bounds for Stochastic Gradient Descent via Localized ε-Covers, NeurIPS, 2022.
>
> **Q4: In the context of Theorem 2, do you give an example of scenario when $\beta\|\mathbf q\|N\rightarrow 0$**
>
> A4:  Thank you for the question. Theorem 2 states that, in the i.i.d. setting, a sufficient condition for the discrepancy term $disc(\mathbf q)$ to vanish asymptotically is $\beta \|\mathbf q\|N->0$. This condition is easy to satisfy under standard choices of the training weights. For example, if we take uniform weights $q_i	=1/N$ for all samples, then $\|\mathbf {q}\| N=N^{1/2}$, and thus the requirement becomes simply $\beta=o(N^{−1/2})$. Our other results (Thm. 1, Cor. 1, and Cor. 2) show that such a decay rate for $\beta$ is achievable under multiple concrete regimes. For instance, Corollary 1 implies that $\beta=o(N^{−1/2})$ holds whenever $\zeta_1+\zeta_2 \gamma/(1+\gamma)<1/2$, where $\zeta_1$ and $\zeta_2$  characterize the growth rates of the batch size and the iteration count $Q$, and $\gamma$ is the Lipschitz-smoothness parameter of the loss.
>
> In the revision, we will explicitly connect Theorem 2 to such concrete and feasible regimes so that readers can clearly see when the condition is satisfied in practice (**please see line 365-370**).

---

> ### Author Response · Authors · 2025-11-21
> **Rebuttal to Reviewer J7sq (3/3)**
>
> **Q5: In table 2, the paper presents the convergence rate, does it not take into account the convergence of training loss ?**
>
> A5: Thank you for the question. The convergence rates reported in Table 2 do not describe the convergence of the training loss itself. Instead, they characterize the rate at which the generalization gap (i.e., the difference between the expected risk and the empirical training loss) vanishes. These results hold regardless of the specific value of the training loss and therefore apply to any training-loss trajectory produced by SGD-type optimization. This convergence analysis is particularly valuable because the training loss is observable, while the expected risk is not. The results in Table 2 therefore provide meaningful guarantees by quantifying how the known training loss approximates the unknown prediction error.

---

### Meta-Review · Area_Chair_39T6 · 2026-01-04

**Summary:**

This paper develops a structured framework based on algorithmic stability and distributional discrepancy for in-context learning (ICL). The angle is interesting and the paper is clear and well organized. The criticism is mostly on the practical impact, which is often the case for theoretical papers. As an example, some comments argued that most experiments are synthetic and rely on small GPT-2–scale models rather than realistic LLM settings. Another example is about the assumptions on boundedness and Lipschitz smoothness. I personally do not view these identified shortcomings as a deal-breaker, as long as the authors are clear about them in the paper. On the other hand, I did not find any review comments that directly question the main technical contributions. I feel that the negative scores are largely due to worries about practicality rather than major technical issues with the paper. As a result, I feel this paper can be recommended for acceptance, although I'll not full-heartedly champion it due to a lack of ground-breaking results.

**Reviewer Concerns:**

I feel that the authors did a decent job addressing the practicality and the assumption questions. They also did substantially more experimentation to answer reviewers' questions.

**Reviewer Scores:**

I personally doubt that the reviewers will increase their scores much. This is mainly because it is clear that the theoretical flavor is not appreciated by multiple reviewers. This is obvious from their questions -- they didn't digest the details of the theory; rather, they ask questions of "usefulness" of the theorems. It is a valid question, but such questions can be posed for almost all theory papers, and it is not reasonable to ask a paper to develop a theory that can match practical LLMs.

---

### Decision · Program_Chairs · 2026-01-26

Accept (Poster)